# Adaptive ionic liquid polymer microwave modulation surface with reprogrammable dielectric properties

Qichao Dong [1,2], Zhehui Wang[3], Hanyu Qiu[3,4], Xiaofeng Gong[5], Huying Yan[1], Zengyong Chu [5,6] ✉, Tao Luo [3] ✉, Haipeng Lu [1,2] ✉ & Longjiang Deng [1,2]

Adaptive microwave surfaces have the capability to dynamically adjust their electromagnetic transmission to meet specific needs, offering significant potential for efficient integration and flexible use in reconfigurable communication systems. In this work, we utilize temperature induced break and reconstruction of hydrogen bonds to drive the orientational motion and charge mobility of the ionic liquid $[EtA^+][NO_3^-]$ in the poly-2-hydroxyethylacrylate, resulting controllable modulation of dielectric properties at microwave frequencies. Building on this mechanism, we applied machine learning algorithms to establish correlations between temperature, ionic liquid concentration, and dielectric constant, enabling the design of a reprogrammable dielectric microwave modulation surface. For example, the 2 mm-thick switchable microwave absorbing surfaces fabricated here can operate in two distinct modes during the temperature transition from low to high, namely, off-to-on and on-to-off. The corresponding tunable effective absorption bandwidths and reflection loss values reach 5.69 GHz, −6.04 dB to −46.21 dB, and 5.34 GHz, −50.48 dB to −6.47 dB, respectively. Using the developed active surface, we also demonstrate various device architectures fabricated by three-dimensional printing, including pixelated surfaces and self-sensing functionalities, which provide valuable guidance for the development of next-generation intelligent electromagnetic devices.

In recent years, the increasing prevalence of complex and dynamic environments in practical electromagnetic wave technology applications has created an urgent demand for active electromagnetic devices capable of real-time adjustment of their electromagnetic properties[1–7]. Typically, radar and communication systems primarily use passive surfaces to guide microwaves and employ dielectric or magnetic materials to control the transmission efficiency of microwaves in free space, such as some of the passive surfaces we previously developed based on dielectric loss and magnetic loss mechanisms[8–11]. Currently, manufacturing active surfaces capable of controlling microwave

[1]National Engineering Research Center of Electromagnetic Radiation Control Materials, School of Electronic Science and Engineering, University of Electronic Science and Technology of China, Chengdu 611731, China. [2]Key Laboratory of Multispectral Absorbing Materials and Structures of Ministry of Education, School of Electronic Science and Engineering, University of Electronic Science and Technology of China, Chengdu 611731, China. [3]Institute of High Performance Computing (IHPC), Agency for Science, Technology and Research (A*STAR), 1 Fusionopolis Way, #16-16 Connexis, Singapore 138632, Republic of Singapore. [4]School of Science and Engineering, The Chinese University of HongKong, Shenzhen 518172, China. [5]College of Science, National University of Defense Technology, Changsha 410073, P. R. China. [6]Science and Technology on Advanced Ceramic Fibers and Composites Laboratory, College of Aerospace Science and Engineering, National University of Defense Technology, Changsha 410073, P. R. China. ✉e-mail: chuzy@nudt.edu.cn; luo_tao@a-star.edu.sg; tluo001@e.ntu.edu.sg; luhaipeng@uestc.edu.cn

communications to meet the requirements of various electromagnetic environments remains a significant challenge. To tackle this issue, researchers have proposed several strategies for fabricating an active microwave modulation surface. Under external stimuli such as voltage, heat, or light, certain surfaces can undergo intrinsic changes in their electromagnetic properties, enabling dynamic and real-time control of microwave signals. For instance, arrayed active circuit elements assembled with passive metallic structures have been investigated[1,3–5,12]. Since external stimuli primarily affect the micron-scale circuit components within the array, the fabrication of large-area surfaces and the microwave control effects are limited. Other studies have typically focused on tuning the permittivity of materials, for example, by utilizing the polarization state of ferroelectric materials[13–15] or the conductivity of phase-change materials[2,16–20], which enables dynamic changes in microwave transmission properties. However, the limited tunability of dielectric properties in the micro-wave frequency range has hindered their prospects for application in active microwave surfaces.

The real ($\varepsilon'$) and imaginary ($\varepsilon''$) parts of the permittivity are critical for microwave modulation. $\varepsilon'$ determines impedance matching at the surface, while $\varepsilon''$ governs attenuation within the material, and an appropriate balance between them reduces interfacial reflection and enhances internal loss[21]. Ionic liquid (IL), well known as a carrier migration system, has been used in polymers as a material for tuning dielectric properties[22,23]. However, to the best of our knowledge, their tunability at microwave frequencies and the underlying mechanism have yet to be demonstrated. In this frequency range, dielectric modulation primarily arises from the orientational motion of dipoles, as well as from conductivity resulting from charge carrier migration[24,25]. Interestingly, in the ionic liquid–polymer (IL-P) system designed in this work, the cation [EtA$^+$] and anion [NO$_3^-$] not only serve as charge carriers but also function as typical dipoles. In addition, a large hydrogen-bond network is formed between IL and IL and between IL and the polymer (poly-2-hydroxyethyl-acrylate, PHEA). The break and reconstruction of these hydrogen bonds under temperature stimulation inevitably lead to significant changes in the orientation and migration motion of IL, thereby causing a drastic transformation in the $\varepsilon'$ and $\varepsilon''$. Unlike previous studies that only focused on the control mechanism of microwave signals driven by $\varepsilon''$[2,17]. Here, we propose the fabrication of active microwave modulation surfaces by adjusting the $\varepsilon'$ and $\varepsilon''$ of the IL-P system via temperature stimuli, while optimizing the modulation performance by IL concentration. Through machine learning algorithms, the corresponding relationship between IL concentration, stimulus temperature, and the electromagnetic parameters of active IL-P was obtained, where $\varepsilon'$ and $\varepsilon''$ were modeled using exact Gaussian process regression (EGPR), while the real ($\mu'$) and imaginary ($\mu''$) parts of permeability were predicted using gradient boosted decision trees (GBDT). This machine learning-driven reprogrammable dielectric design concept provides new solutions for the electromagnetic applications of IL-P, including active stealth technology, reconfigurable microwave antennas, and high-security microwave communication systems. For example, using this method, 2 mm-thick switchable microwave absorbing surfaces composed of IL-P with two different IL concentrations were designed, which exhibited distinct switching modes during the temperature increase process. In the off-to-on mode, $\varepsilon'$ increases from 4.79 to 6.43, and $\varepsilon''$ increases from 1.06 to 2.78 at 14.76 GHz. The tunable effective absorption bandwidth ($\Delta$EAB) reaches up to 5.69 GHz, with reflection loss (RL) values ranging from −6.04 dB to −46.21 dB. In contrast, in the on-to-off mode, $\varepsilon'$ ranges from 7.73 to 11.20, and $\varepsilon''$ ranges from 3.14 to 6.98 at 13.63 GHz, exhibiting a $\Delta$EAB of up to 5.34 GHz and RL values ranging from −50.48 dB to −6.47 dB. In addition, due to the microwave wavelength being on the centimeter scale, the material fabrication and structural design of large-area, actively tunable microwave surfaces face significant challenges[1,26]. We have overcome these challenges by adopting

a light-activated polymerization-based IL-P 3D printing technique, which enables easy fabrication of various large-area designs and holds promise for precise control of multi-pixel active reflectance modes. In addition to the active electromagnetic performance modulation, IL-P exhibits high sensitivity to external environmental stimuli. With this highly perceptive tactile capability, various stimulus signals detected at different surface locations can be encoded and transmitted to the data processing center, where a convolutional neural network (CNN) transformer hybrid can further identify the location and magnitude of the surface stimuli[27,28]. This endows the surface with self-diagnostic capability and provides a robust platform for the realization of large-area adaptive microwave surfaces.

## Results

### Electromagnetic regulation of IL-P

Figure 1a presents a schematic illustration of the electromagnetic property modulation of IL-P. The regulation of electromagnetic properties is based on a hydrogen-bonded network formed between the IL and the polymer matrix under thermal stimulation. Applying a temperature stimulus (30–130 °C) to IL-P induces break and reconstruction of hydrogen bonds between the IL and PHEA, resulting in a transition in the dynamic state of the IL. Freed from hydrogen-bond constraints, the IL exhibits tunable orientational motion and generates high-mobility charge carriers, resulting in changes in conductivity and dynamic permittivity regulation.

As shown in Fig. 1b and Supplementary Fig. 1, after being 3D printed on a digital light processing printer, the IL-P was subjected to 1000 cycles of 30–130 °C temperature thermal shock in a thermal shock chamber. Firstly, phase transition mechanisms are commonly involved in the regulation of electromagnetic parameters[2,16–19], so it is necessary to exclude this factor by thermogravimetric and differential scanning calorimetry (TG-DSC). IL-P exhibits excellent thermal stability in the range of 30–130 °C, with no chemical decomposition or physical phase transition observed (Supplementary Fig. 2). Secondly, although some previous studies have attempted to control electromagnetic wave transmission by adjusting the thickness of active surfaces[29,30], our kelvin probe force microscopy (KPFM) results clearly rule out the possibility that IL-P is influenced by this factor. The thickness of the IL-P film measured by KPFM remains unchanged before and after the temperature change (Fig. 1c). From the uniform distribution of surface potential across the IL-P surface, it can be seen that no IL enrichment occurs in the microscopic regions after hydrogen bond break, thus excluding the impact of interfacial polarization effects on the electromagnetic waves[31,32]. It is worth noting that the surface potential of IL-P increases from 140 to 180 μV (Fig. 1d), indicating a reduction in the surface work function at higher temperatures[33]. This further suggests an adjustment of the energy level structure, with an increase in the concentration of free charge carriers and enhanced polarization capabilities.

According to density functional theory (DFT), as shown by the differential charge density in Fig. 1e, these phenomena observed by KPFM after heating IL-P can be attributed to the break of the extensive hydrogen-bonding network among the cation [EtA$^+$], anion [NO$_3^-$], and PHEA. The break and reconstruction of hydrogen bonds under thermal stimulation control the free charge carrier concentration and polarization capability of IL in the system. Moreover, Fig. 1f shows that statistical analysis of the average hydrogen bond density from molecular dynamics simulations (MDS) reveals that increasing IL concentration decreases the hydrogen bond density between IL and PHEA, but increases it between [EtA$^+$] and [NO$_3^-$]. This indicates that the number of available binding sites provided by PHEA for the IL decreases, resulting in a higher proportion of free IL in the system. It is evident that IL concentration inevitably affects the regulation of dielectric properties. It is noteworthy that the hydrogen bond density between PHEA chains remains stable with increasing IL concentration,

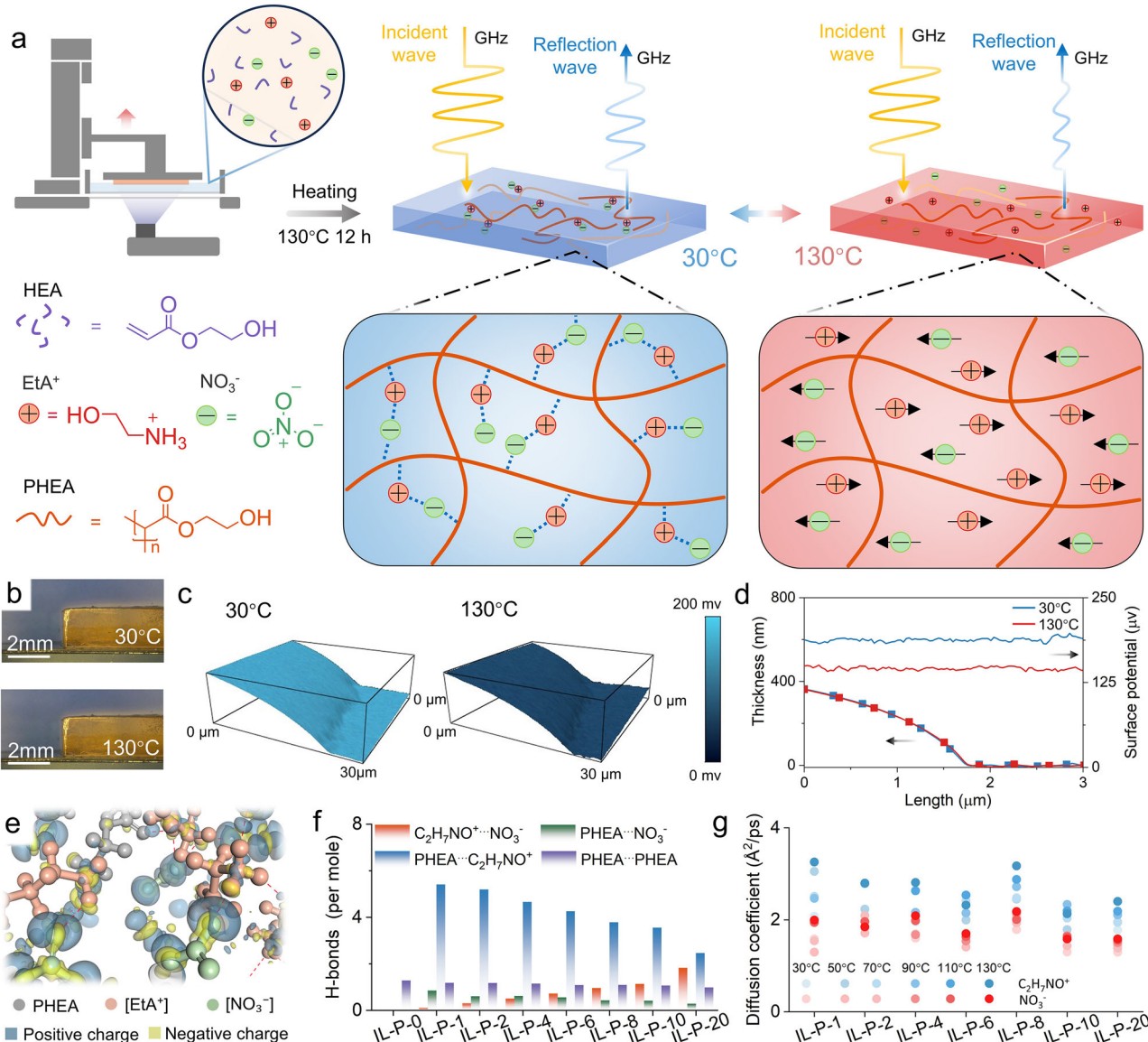

**Fig. 1 | The mechanism of electromagnetic regulation in IL-P. a** Schematic illustration of electromagnetic property modulation based on IL-P. **b** Macroscopic morphology of IL-P at 30 and 130 °C. **c** KPFM of IL-P films at 30 and 130 °C. **d** Thickness and surface potential of IL-P films at 30 and 130 °C. **e** Hydrogen-bond network and differential charge density in the IL-P system, isosurface value: 0.02 e/ Å³. **f** Hydrogen bond densities between different components in IL-P systems with various IL concentrations. **g** Molecular dynamics diffusion coefficients of IL under different temperature fields.

indicating that the IL is unlikely to affect the relaxation motion of PHEA. These variations in IL concentration are beneficial for the design of electromagnetic devices with diverse tunability requirements. Moreover, as shown in Fig. 1g, under different temperature fields in the NVT ensemble, the diffusion coefficient of IL in PHEA increases by more than twofold after hydrogen bond break. This significant enhancement further confirms the generation of a large number of free dipoles and charge carriers[34], which is sufficient to induce dramatic changes in the electromagnetic properties of IL-P. The conclusion will be further validated in the subsequent IL-P-2.

## IL dynamics and dielectric behavior

To more accurately elucidate the mechanism of electromagnetic property switching controlled by the IL-P, a representative IL-P-2 sample was selected for analysis. In situ Fourier transform infrared spectroscopy (FTIR) measurements during cycles between 30 and 130 °C, as shown in Fig. 2a, reveal that the infrared transmission peaks of both −OH and −NH₃ increase and decrease synchronously,

indicating that the hydrogen-bonding network in IL-P undergoes break and reconstruction[35]. The strong autopeaks associated with −OH and −NH₃ in the two-dimensional (2D) synchronous FTIR spectra (Fig. 2b) further confirm that disruption of the hydrogen-bonding network is the most significant change in IL-P-2 under thermal stimulation. The autopeak of CH₂ suggests that the disruption of the hydrogen-bonding network may induce changes in the segmental motion of PHEA molecular chains, thereby affecting the polarization relaxation behavior of PHEA. More importantly, according to Noda's rule, analysis of the 2D synchronous and asynchronous FTIR spectra in Fig. 2b, c reveals the sequential order of spectral intensity changes under thermal stimulation: OH > N−O > −NH₃ > C=O and C−O−C. This indicates that the −OH groups on [EtA⁺] and PHEA contribute most significantly to the break of the hydrogen-bonding network under thermal stimulation, followed by the N−O group on [NO₃⁻], while the C=O and C−O−C groups on PHEA make the weakest contributions, which is consistent with the previous MDS analysis. As shown in Supplementary Fig. 3, the IL content in the PHEA does not affect the contribution to hydrogen

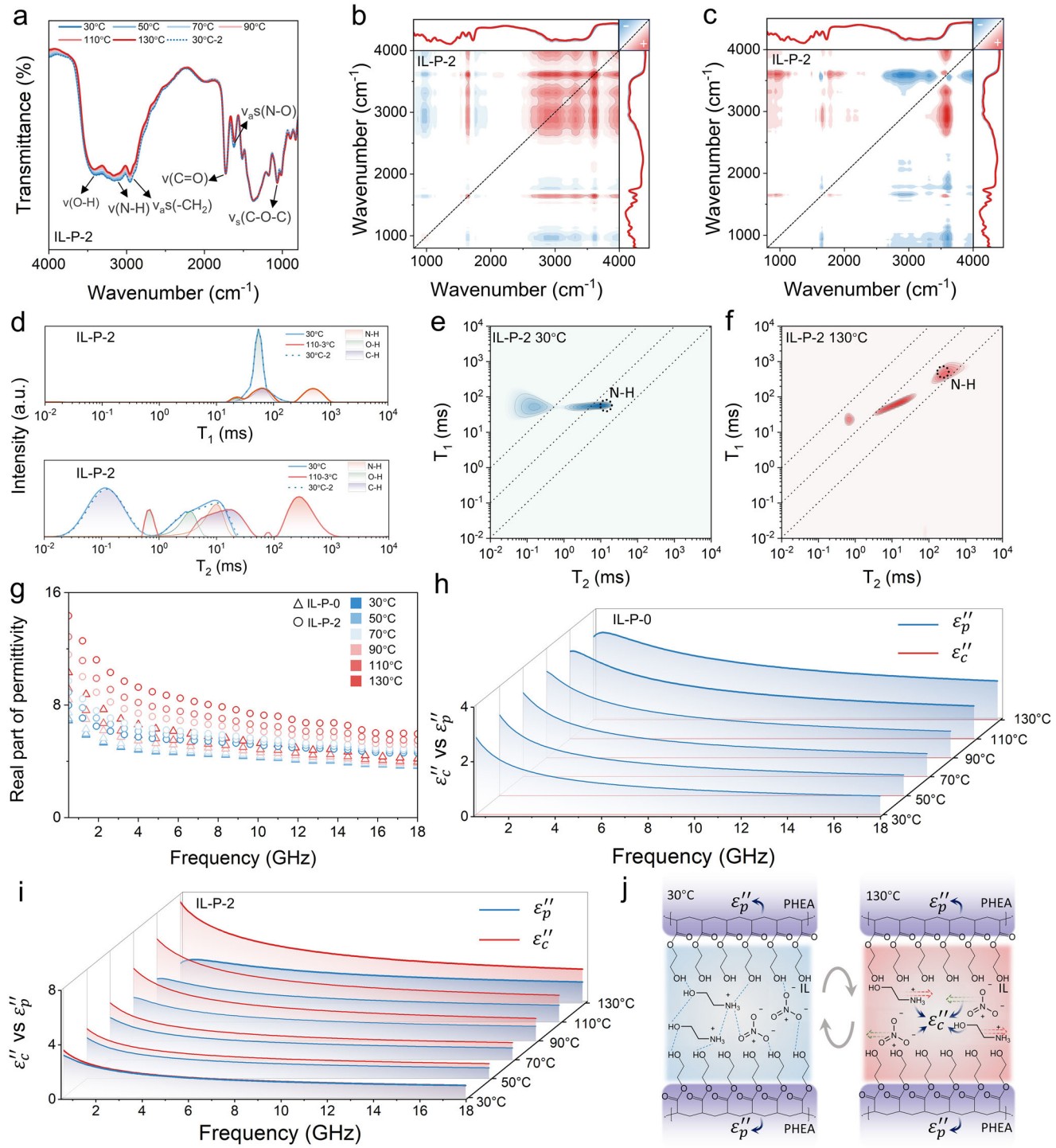

**Fig. 2 | The effect of [EtA⁺][NO₃⁻] mobility in the PHEA on electromagnetic properties. a** In situ FTIR of IL-P-2 under thermal stimulation. **b, c** VT-FTIR synchronous and asynchronous correlation spectroscopy. **d** $T_1$ and $T_2$ spectra of IL-P-2 in low-field NMR under different temperature stimuli. **e, f** 2D low-field $^1$H NMR spectra at 30 and 130 °C. **g** ε′ of IL-P-0 and IL-P-2. **h** Dielectric loss fitting results for IL-P-0. **i** Dielectric loss fitting results for IL-P-2. **j** Mechanism of temperature stimulus on the ε″ of IL-P.

bond break, meaning the mechanism driving the change in electromagnetic properties during the heating process remains the same. Therefore, it can be concluded that the IL content in the IL-P system has a unique and significant impact on the electromagnetic parameters.

Previous MDS indicated that hydrogen bonding drives changes in the migration state of IL within PHEA, and the results of low-field $^1$H NMR measurements further support this mechanism[36]. As shown in

Fig. 2d and Supplementary Fig. 4, the $T_1$ and $T_2$ spectra of the C–H and O–H groups in PEAN for both IL-P-0 and IL-P-2 are unaffected by the presence of the IL. This indicates that the relaxation and polarization behavior of PEAN in IL-P-2 is almost identical to that in IL-P-0. The relaxation motion of the PEAN backbone under thermal stimulation remains in the range of $10^1$–$10^2$ ms, constrained by the cross-linked network. However, the opposite changes observed in the $T_2$ spectra of C–H and O–H indicate that, at high temperatures, the mobility of

hydrophilic −OH groups increases, while hydrophobic C−H groups may undergo local aggregation. This dynamic heterogeneity may influence the dielectric polarization behavior of PEAN and facilitate the migration of IL within the system. A clear comparison of the 2D low-field NMR spectra in Fig. 2e, f reveals that, in IL-P-2, the $T_1$ (spin−spin) and $T_2$ (spin−lattice) relaxation times associated with the N−H from [EtA$^+$] exhibit an order-of-magnitude increase under thermal stimulation. The increase in $T_1$ from 53 ms to 530 ms indicates that the break of hydrogen bonds weakens the intermolecular interactions, causing the energy exchange process between [EtA$^+$] and PHEA to slow down. At the same time, the increase in temperature leads to greater local heterogeneity in the PHEA, and the enhanced chain mobility may cause the local environment around the −NH$_3$ group on [EtA$^+$] to become more relaxed, allowing for freer movement. This reduces spin−spin interactions, resulting in an increase in $T_2$ from 9 ms to 2616 ms[37]. These drastic changes in the dynamic state of the IL, which serves as both dipoles and charge carriers in the system, are expected to have a significant impact on the electromagnetic properties at microwave frequencies[23,37].

The permittivity (ε′, ε″) and permeability (μ′, μ″) of the material were measured at different temperatures across 2–18 GHz using a vector network analyzer (VNA) to evaluate its dielectric and magnetic responses. The dielectric relaxation behavior of the polymer in IL-P was assessed using an impedance analyzer in the frequency range of 10$^7$ to $5 \times 10^8$ Hz. Compared to previous reports, the significant increase in ε′ observed in the IL-P system within this frequency range is unique[2,17,18,21]. For IL-P-0, composed of PHEA, the enhancement of the autopeak in 2D synchronous IR and the dynamic heterogeneity observed in low-field NMR indicate an increase in its relaxation polarization ability, which corresponds to the observed increase in ε′, consistent with the test results, as shown in Fig. 2g. In IL-P-2, KPFM, in situ FTIR, and low-field $^1$H NMR have demonstrated that the gradual weakening of hydrogen bonds under temperature stimulation leads to a decrease in the diffusion energy barrier of the IL, an increase in the number of dipoles participating in orientational motion, and enhanced mobility of charge carriers controlling conductivity, all of which drive the increase in ε′. Compared to the single relaxation polarization mechanism of the polymer PHEA in IL-P-0, the change in ε′ driven by the IL is more pronounced. Furthermore, under this modulation mechanism, the change in ε′ becomes more significant with an increasing amount of IL, as shown in Supplementary Figs. 5 and 6.

The ε″ of IL-P in the range of 0.5 to 18 GHz is shown in Supplementary Fig. 7. As the hydrogen bonds break and the mobility of the IL increases, ε″ shows a positive correlation with temperature. The mechanism of ε″ is typically more complex than that of ε′, as it involves multiple dynamic processes, including charge carrier migration, dipole relaxation, and interfacial effects[2,8], which can be evaluated using the Havriliak–Negami (H−N) model[38]. Firstly, PHEA contains almost no freely migrating electronic bands or ions. Therefore, ε″ in IL-P-0 primarily arises from polarization relaxation (ε″$_p$), as shown by the fitting results in Fig. 2h. The shift of the polarization relaxation peak observed in the 10$^7$–$5 \times 10^8$ Hz range strongly supports this notion, as shown in Supplementary Fig. 8A. In addition, combined low-field $^1$H NMR, in situ FTIR, and MDS results show that the presence of the IL does not affect the relaxation polarization behavior of PHEA. The similarity of the relaxation peaks of ε″$_p$ from PHEA in IL-P-0 and IL-P-1 at the characteristic relaxation frequency, as shown in Supplementary Fig. 8, further supports this observation. Based on this, through fitting with the H−N model, we can obtain the transition in the relaxation time of PHEA in IL-P and the conductivity change induced by the IL, as shown in Supplementary Fig. 9. Specifically, the conductivity of IL-P-2 increases from $2.8 \times 10^{-3}$ S/m at 30 °C to $6.2 \times 10^{-2}$ S/m at 130 °C. As demonstrated by KPFM, low-field $^1$H NMR, and MDS simulation results, this dramatic change is attributed to the increased mobility of the IL following hydrogen bond break, which leads to a higher concentration

of charge carriers in the IL-P-2 system. The sharp increase in ε″$_c$ resulting from the enhancement of ionic conductivity is the primary mechanism influencing ε″ of IL-P-2, as shown in Fig. 2i. According to the mechanism shown in Fig. 2j, along with the conclusions from MDS and in situ FTIR, the change in ε″ of IL-P under thermal stimulation is theoretically more pronounced with increasing IL content. The experimental (Supplementary Figs. 7 and 8) and fitting results (Supplementary Fig. 10) align with the expected outcomes, further supporting the universality of this modulation mechanism.

During the heating process, due to the significant contribution of ε″$_c$, ε″ dominates and increases at a faster rate than ε′. As a result, the dielectric loss tangent (tan $\delta_e$) of IL-P shows an increasing trend, as shown in Supplementary Fig. 11. Interestingly, the IL-P structure lacks elements or configurations that support magnetic behavior, and consequently, the magnetic permeability, which includes μ′ (Supplementary Fig. 12), μ″ (Supplementary Fig. 13), and the tan $\delta_\mu$ (magnetic loss tangent) (Supplementary Fig. 14), is minimally affected by temperature. Based on this, the modulation of IL-P impedance matching and energy loss properties can be easily achieved through changes in dielectric characteristics. In summary, the tunable dielectric properties arise from the dynamic transformation of the IL driven by hydrogen bonds under temperature stimuli. Clearly, the IL concentration has a decisive impact on the system's dielectric properties. Theoretically, controlling the IL concentration can easily enable the development of active devices that meet various electromagnetic control needs at microwave frequencies.

## Reprogrammable dielectric microwave surface

Based on the understanding of the IL-P electromagnetic property modulation mechanism under temperature stimuli, it is evident that the significant dielectric modulation range induced by IL concentration can be utilized to achieve the desired actively switchable microwave absorbing surfaces. Therefore, we constructed a structure consisting of an IL-P layer and a flat metal surface at the bottom to evaluate the switchable microwave absorption performance. The main sources of RL rely on impedance matching achieved through the ε′, and energy loss mechanisms caused by ε″[2,10].

As shown in Fig. 3a, b, we compared the dielectric response characteristics at the core microwave frequency of 10 GHz. With the further increase in IL content, the hydrogen bonding sites provided by PHEA become saturated, and a large amount of free IL appears in the system. Under low-temperature conditions, both ε′ and ε″ increase gradually with the rise in IL concentration. Moreover, the introduction of high-concentration IL brings more controllable dipoles and charge carriers, enhancing the modulation range of ε′ and ε″, accompanied by an increase in tan $\delta_e$ (see Fig. 3c). Interestingly, due to the stability of the magnetic permeability (Fig. 3d), the transmission behavior almost entirely depends on the modulation of the aforementioned dielectric properties, making the IL concentration a key factor for the electromagnetic wave control performance[2,10]. Therefore, based on the trend of these electromagnetic parameters changing with temperature, it can be concluded that the attenuation coefficient is positively correlated with both temperature and the IL concentration in the polymer (Fig. 3d). This indicates that higher temperatures lead to greater attenuation of microwaves during propagation through the IL-P.

Building on the aforementioned electromagnetic response mechanism, we further introduced machine learning algorithms to achieve high-precision modeling and prediction of the electromagnetic parameters of the IL-P system under different temperature conditions, As shown in Fig. 3e and Supplementary Fig. 15. Based on the measured data (Supplementary Figs. 5, 7, 12, and 13), significant differences in the variation trends of permittivity and permeability can be observed. In response to these differences, we used the EGPR model to predict ε′ and ε″, and the GBDT model to predict μ′ and μ″, thereby systematically revealing the coupled regulatory relationship between

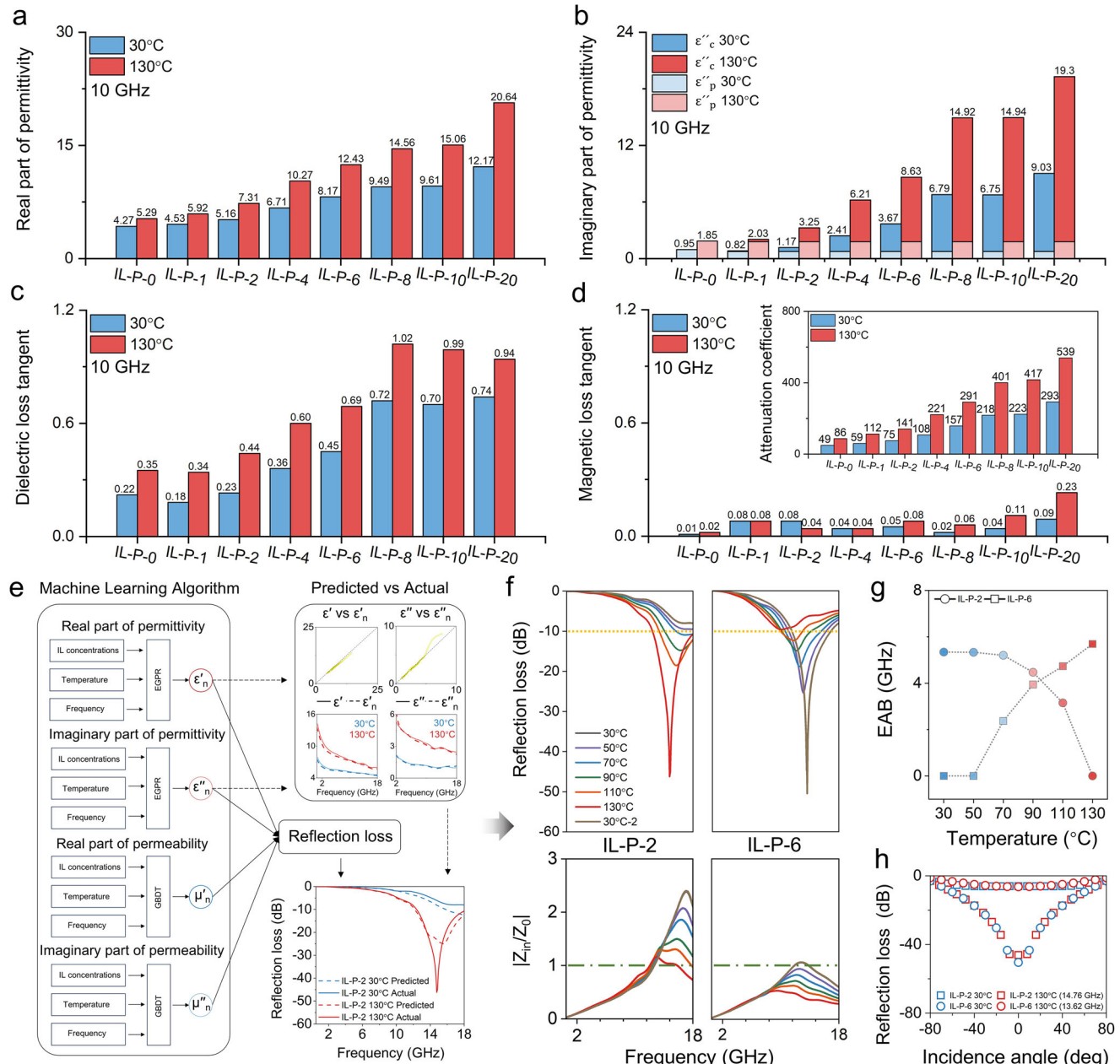

**Fig. 3 | Reprogrammable microwave absorbing surfaces composed of IL-P with different IL concentrations. a–c** $\varepsilon'$, $\varepsilon''$, and tan $\delta_\varepsilon$ of IL-P at 10 GHz under different temperature stimuli. **d** tan $\delta_\mu$ of IL-P at 10 GHz under different temperature stimuli, with the attenuation coefficient shown in the upper right corner. **e** Modeling and prediction of electromagnetic parameters and RL under multivariable coupling conditions based on machine learning algorithms, where $\varepsilon'_n$, $\varepsilon''_n$, $\mu'_n$, $\mu'_n$, represent the predicted values(dashed lines: predicted values, solid lines: actual values). **f** The matching degree of the corresponding wave impedance with free space impedance ($Z_0 = 377$), $Z_{in}/Z_0$, and the normal direction RL of the 2 mm-thick microwave modulation surfaces composed of IL-P-2 and IL-P-6 in the 0.5–18 GHz range under different temperature stimuli are calculated based on the measured electromagnetic parameters. **g** ΔEAB of IL-P under different temperature stimuli. **h** RL of the switchable microwave absorbing surface for microwaves at incident angles ranging from −80° to 80°.

IL concentration, temperature stimulation, electromagnetic parameters, and RL. Using this model and data foundation, we established a target-prediction-driven active control loop, achieving programmable design of microwave absorption performance, and successfully constructed two switchable state microwave absorbing surfaces. Incredibly, the switchable microwave absorbing surface with an ionic liquid concentration close to that of IL-P-2 and a thickness of only 2 mm could exhibit the off-to-on mode (Supplementary Fig. 16), while another with an ionic liquid concentration close to that of IL-P-6 could also exhibit the on-to-off mode (Supplementary Fig. 17). For IL-P-2, in addition to the impact of dielectric changes on impedance matching, the sharp increase in attenuation coefficient with temperature rise is

the key factor driving the off-to-on transition of microwave absorption capability. At 14.76 GHz, the RL value increases from −6.04 dB to −46.21 dB, and it possesses an ΔEAB of 5.69 GHz, as shown in Fig. 3f, g. In addition to the off-to-on mode microwave surface driven by attenuation coefficient, IL-P-6, which utilizes impedance mismatch changes induced by dielectric properties, can achieve the on-to-off mode switching. Due to the increase in permittivity caused by the rise in temperature, the $|Z_{in}/Z_0|$ of IL-P-6 deteriorates from 1 to 0.5. Due to impedance mismatch caused by the increase in temperature, electromagnetic waves are reflected at the microwave absorbing surface and fail to couple effectively into the material. As a result, RL deteriorates from −50.48 dB to −6.47 dB, and the ΔEAB gradually decreases from

5.34 GHz to 0, at 13.63 GHz. In addition, the repeatability of the electromagnetic switching in active microwave absorbing surfaces is crucial for practical applications. IL-P-2 was selected as a representative sample and demonstrated extremely stable electromagnetic control characteristics through high-low temperature cycling tests (Supplementary Fig. 18). Across a wider range of incident angles, these devices still maintain significant control performance, as shown in Fig. 3h.

The designable effective modulation bandwidth is an important requirement for reconfigurable microwave-absorbing surfaces. Firstly, impedance matching, achieved by adjusting the material thickness, can easily shift the effective modulation bandwidth (Supplementary Fig. 19). Secondly, the metasurface structure enables the absorption bandwidth of conventional passive microwave absorbing surfaces to be modified[1,5,12,39]. Based on this theoretical foundation and design concept, the effective modulation bandwidth of IL-P-based active microwave absorbing surfaces was tailored by designing metasurface structures or overlaying frequency-selective surfaces (Supplementary Fig. 20). This strategy broadens the effective modulation bandwidth, thereby improving adaptability and performance under complex electromagnetic conditions. In addition, IL-P also exhibits the potential to control microwave transmission (see Supplementary Fig. 21), further showcasing the ability of the IL-P to manipulate electromagnetic waves.

By controlling the IL in IL-P at the molecular level, dynamic dielectric properties in the microwave frequency range are achieved, showing superior modulation capabilities compared to existing methods, as shown in Supplementary Table 1. Building on this, the active microwave-absorbing surfaces composed of IL-P surpass most of the current reports[2,4,6,19–21]. The above IL-P, based on a machine learning-driven reprogrammable dielectric design concept, provides new solutions for a range of cutting-edge electromagnetic applications, including active stealth technology, reconfigurable microwave antennas, and high-security microwave communication systems.

## Switchable microwave absorbing surfaces
By dynamically modulating the on and off states of multiple active unit cells on a microwave absorbing surface, it is possible to form reflectivity patterns on a multi-pixel adaptive surface, enabling a wide range of possibilities for active camouflage systems. This concept has already been demonstrated at terahertz frequencies[7]. Here, we demonstrated these strategies using a square pixelated surface composed of IL-P. A single surface is made up of a 2 mm-thick flexible 10 cm² IL-P-2 (Fig. 4a) and a metal–polyimide conductive heating film, with testing shown in Supplementary Fig. 22a, b. Figure 4b shows the microwave reflectance maps of a single pixel at various temperatures. The strong reflection at 30 °C is weakened at 130 °C due to the increased microwave absorption by the surface. We then explored a 3 × 3 pixel configuration for imaging under different temperature conditions (Fig. 4c, d and Supplementary Fig. 22c–f). Controlling the reflectance pattern through this large-area, multi-pixel switchable microwave absorbing surface creates various possibilities for active camouflage systems.

In addition, the radar cross section (RCS) is also a key indicator for evaluating the potential of switchable microwave absorbing surfaces in active camouflage systems[2,21,31]. The active surfaces of IL-P-2 and IL-P-6, with a thickness of 2 mm, are coated on a 10 × 10 cm metal surface. At frequencies of 14.76 GHz and 13.63 GHz, the RCS results of the switchable microwave absorbing surface show significant differences at 30 °C and 130 °C as the direction angle of the electromagnetic wave (θ value) changes from 0° to 180° (Fig. 4e). Specifically, IL-P-2 exhibits excellent RCS values at 130 °C across different electromagnetic wave direction angles, significantly lower than the RCS at 30 °C. This phenomenon strongly confirms the active camouflage performance of IL-P-2, switching from off-to-on with the increase in temperature (Fig. 4f). As expected, the RCS of IL-P-6 indicates that with the increase in temperature, the active camouflage performance transitions to an on-to-off mode (Fig. 4g). Notably, to further verify its application potential in real-world scenarios, switchable RCS simulations were conducted on an aircraft, demonstrating excellent RCS regulation capability (Supplementary Fig. 23). This RCS-controllable phenomenon strongly supports the potential of IL-P as a next-generation active camouflage surface, with its control capability far exceeding that of existing reports[1–3,12].

## Structural health monitoring
The integration of microwave absorbing surfaces with intelligent sensing technology can significantly expand the potential application dimensions of active camouflage systems[1,4]. When the skin is impacted or injured, the biological immune system can immediately identify and locate the affected area, enabling further treatment[40]. In a similar manner, active microwave absorbing surfaces should also have the capability for real-time and precise detection of structural health, as they may face various types of structural damage such as impacts, aging-related cracks, and other defects[9,41]. The piezoelectric sensing performance of IL-P has been widely used as a self-sensing material, where the increase in IL concentration is key to enhancing sensing performance. However, the results in Fig. 5a show that the trend of a sharp increase in IL-P sensing performance begins to level off when the IL concentration reaches that of IL-P-2. When the inherent properties of materials reach their limits, microstructures can play a revolutionary role in high-performance sensors[39,42,43]. To further improve IL-P sensing performance, we fabricated two types of sensors with microstructures, square-cone IL-P-2-C and cone-shaped IL-P-2-T, using 3D printing (Fig. 5b and Supplementary Fig. 24). Among them, IL-P-2-T-2 not only accurately detects changes in environmental pressure (Fig. 5c), but also exhibits high sensitivity (Fig. 5d) and excellent repeatability (Fig. 5e). On this basis, a piezoelectric sensor network integrated with IL-P-2-T-2 was constructed, dividing the adaptive surface into several regions, as shown in Supplementary Fig. 25. When a local impact occurs, each piezoelectric sensing unit detects a distinct signal due to the different deformations generated across the adaptive surface. In this way, the CNN-Transformer hybrid learns and calculates according to the corresponding impact regions, establishing a correlation between the impact location and the sensor network response (Fig. 5f), which enables 100% identification of the impact region as well as detection of weak impacts with intensities below 10 g (Fig. 5g). This self-sensing capability offers great potential for realizing adaptive stealth in active microwave absorbing surfaces.

## Discussion
In summary, this study proposes and validates an adaptive microwave surface made from flexible IL-P materials. The research demonstrates that IL-P-based devices can function as actively tunable microwave functional units, with their high-mobility charge carriers and dipole orientation response behaviors dynamically adjusted through temperature-controlled hydrogen bonds. This results in a relatively high tunable dielectric constant in the microwave frequency range, and this tunability can be systematically controlled by the IL concentration within the IL-P system. Building on this, we utilized machine learning models to precisely model and optimize the nonlinear relationships of dielectric parameters under multivariable coupling conditions, achieving the design of a reprogrammable microwave absorbing surface that can switch between off-to-on and on-to-off modes. Furthermore, we preliminarily constructed and validated a variety of integrated device architectures based on this active absorption surface. Their exceptional electromagnetic control performance effectively extends the tunable frequency range and application dimensions of the IL-P system. Furthermore, exploiting the outstanding self-sensing capability of the IL-P material together with an integrated CNN–Transformer hybrid model, we developed an environmentally responsive intelligent active camouflage system. The

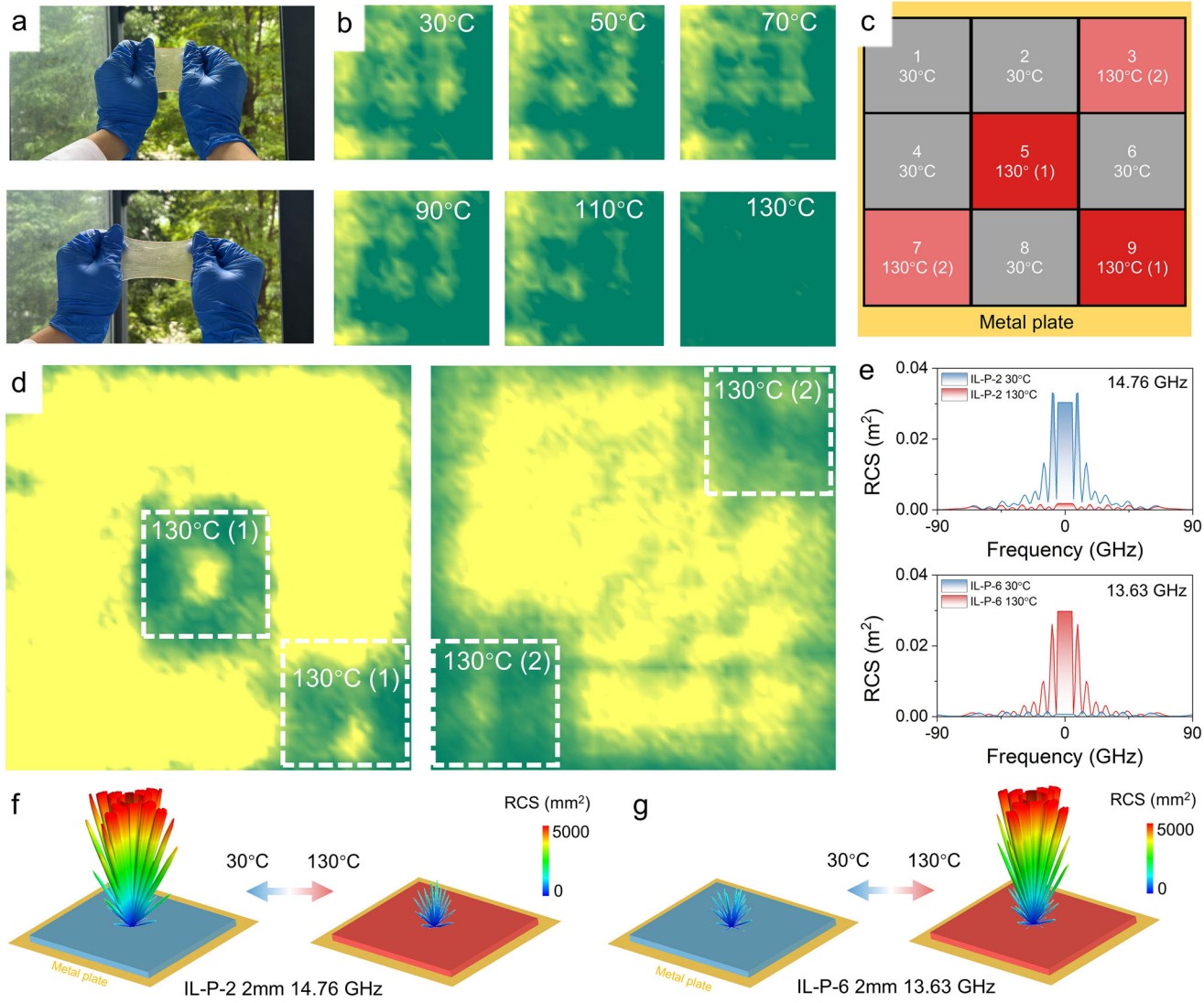

**Fig. 4 | Large-area switchable microwave modulation surfaces. a** Demonstration of large-area flexible IL-P. **b** Far-field scanning microwave reflection images of a single 2 mm-thick IL-P-2 at different temperatures, with an area of 10 × 10 cm. **c**, **d** Pixelated microwave absorbing surface formed by scanned microwave reflection of a 3 × 3 array of nine units with different temperature configurations. **e**–**g** Single-station RCS of a 2 mm-thick switchable microwave absorbing surface, with an area of 10 × 10 cm.

above research demonstrates that the flexible adaptive microwave surface based on IL-P shows significant potential for application in active camouflage systems, and its reprogrammable electromagnetic control capabilities are also expected to be widely applied in microwave electromagnetic devices such as reconfigurable antennas and adaptive filters.

## Methods
### Materials
Analytical reagent grade 2-hydroxyethyl-acrylate (HEA) and the phenyl bis (2,4,6-trimethylbenzoyl) phosphine oxide (Photoinitiator I819) were purchased from Macklin. AR grade nitric acid and ethanolamine were obtained from Keshi.

### Preparation of IL-P
Nitric acid was slowly added to ethanolamine in a double-layer reaction vessel at −10 °C at a rate of 5 ml/min and mixed until neutral. The mixture was then kept at 130 °C for 5 days to obtain the ionic liquid (IL) [EtA$^+$][NO$_3^-$]. I819 and HEA were mixed at a mass ratio of 3:100 and ultrasonicated in the dark to obtain I819-HEA. IL and I819-HEA were

then mixed at a molar ratio of $x$:10 and used for 3D printing by photopolymerization to produce IL-P-$x$. Subsequently, the samples were kept at 130 °C for 1 day and then subjected to 1000 cycles between 30 °C and 130 °C in a thermal shock chamber.

### Design of microstructure
The 2 mm thick IL-P film is designated as IL-P-PL. IL-P with different IL concentrations was sequentially loaded into a photopolymerization 3D printer for layer-by-layer in situ printing to obtain IL-P-MTS. The periodic structure is 2 mm × 2 mm, with each IL-P layer having a height of 1 mm and a four-sided pyramid at the top. A copper sheet with a radius of 0.9 mm is placed on the IL-P-PL surface, with a period of 2 mm × 2 mm, resulting in IL-P-FSS.

### Fabrication of self-sensing systems
IL-P-$x$ films with a thickness of 500 μm were fabricated by 3D photopolymerization printing. These films were used as sensing materials, and capacitive pressure sensors were fabricated using ITO conductive films (Thickness: 125 nm) as electrodes. The IL-P-$x$ formulation was used to prepare IL-P-$x$-C-$y$ and IL-P-$x$-T-$y$ sensors, with sensing layers of

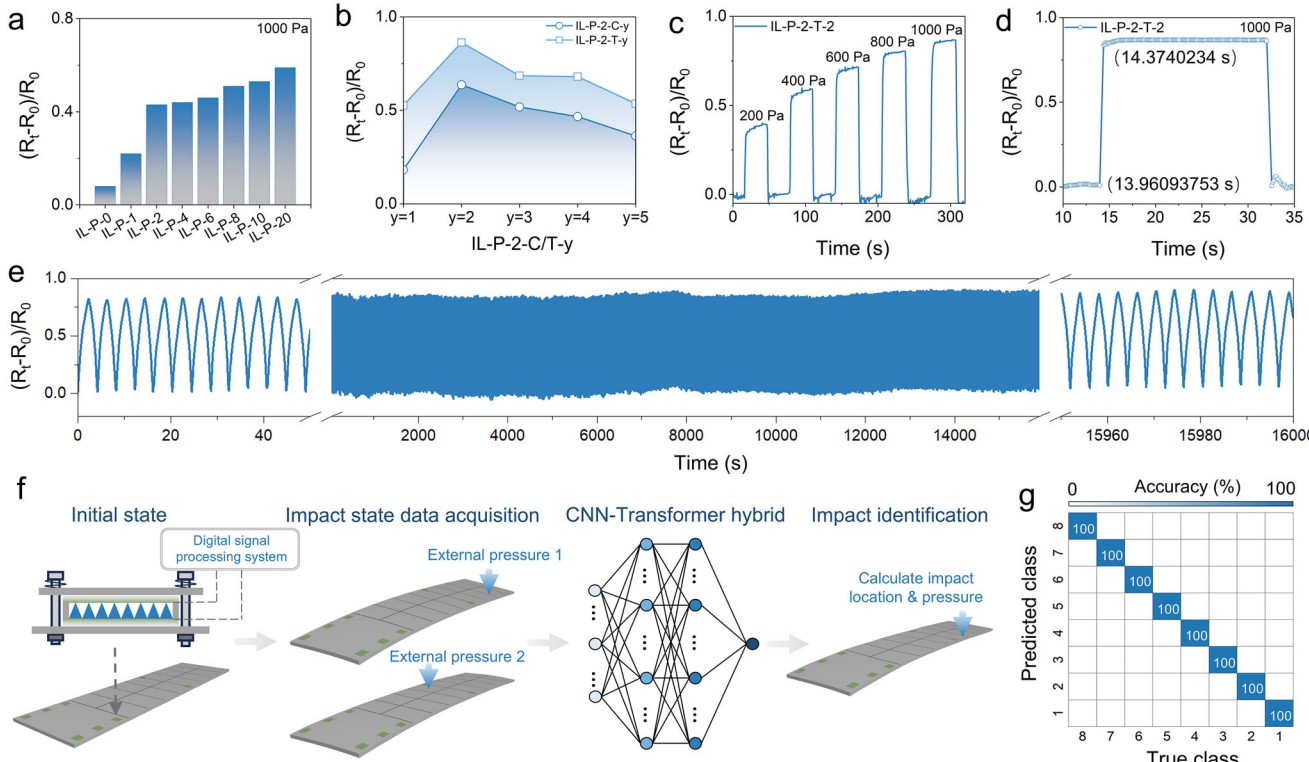

**Fig. 5 | Structural health monitoring enabled by IL-P sensing performance and CNN-transformer hybrid learning technology. a** Sensing capability of IL-P with different IL concentrations. **b** Sensing capability of IL-P-2 with different microstructures. **c**, **d** Dynamic response curves and response/recovery times of IL-P-2-T-2. **e** Repeatability of dynamic response of IL-P-2-T-2 under 1000 Pa pressure. **f** Principle of self-sensing recognition of the IL-P-2-T-2 sensor array based on a CNN-Transformer hybrid model. **g** Confusion matrix for the classification results on the test dataset.

500 μm height and cone or truncated cone geometries, where $y = 1, 2, 3, 4, 5$ represents diameters or side lengths of 250, 500, 1000, 2000, and 4000 μm, respectively. The IL-P sensors with the best performance were selected and mounted on four iron plates compressed by spring screws, and distributed at each recognition site in the self-sensing area.

## Characterization

All variable-temperature measurements were performed in situ, with a 5-min temperature equilibration after each temperature change. KPFM was performed with an OXFORD-MFP-3D Origin+ instrument. Thermogravimetric-differential scanning calorimetry was performed with an STA449F3. FTIR (VT-FTIR) spectra were collected in transmission mode using an INVENIO R. Two-dimensional low-field NMR was measured with a Niumag VTMR20-010V-I system. Electromagnetic parameters were measured in the 2–18 GHz range with a vector network analyzer (VNA) Agilent N5230A using a toroidal ring sample (inner diameter: 3.00 mm, outer diameter: 7.00 mm) and an external temperature control system. Electromagnetic parameters were measured using the Agilent E4991 impedance analyzer over the frequency range of $10^7$ to $5 \times 10^8$ Hz, with the material tested in the form of thin disks with a thickness of approximately 1 mm and a diameter of 7 mm. Conductivity and polarization losses were calculated with Winfit 4.0. Far-field scanning was conducted using a VNA ZNB20, Linbo scanning console, and electromagnetic imaging system. RL and RCS simulations were performed with FEKO.

## DFT and MDS simulations

Density functional theory (DFT) calculations were performed using the CASTEP module in Materials Studio 2025 with the Generalized Gradient Approximation (GGA) and the Perdew–Burke–Ernzerhof (PBE) functional. For the geometry optimization, a plane-wave basis set was employed with an energy cutoff of 630 eV and a convergence criterion of $5.0 \times 10^{-6}$ eV/atom for total energy. The k-point grid of $1 \times 1 \times 1$ was used for electronic structure calculations. The geometry was fully optimized, and the energy was minimized for each structure. Molecular dynamics simulations (MDS) were conducted using the FORCITE module in Materials Studio with the COMPASS force field. The simulation was carried out for a total time of 10 ps with 1000 steps. The NVT ensemble was applied to control the temperature during the simulation. Hydrogen bond statistics were calculated using a custom script to determine the average number of hydrogen bonds throughout the simulation period.

## Data availability

The data generated in this study have been deposited in the figshare repository under accession code: https://doi.org/10.6084/m9.figshare.30774278. All data are available from the corresponding author upon request.

## Code availability

Scripts used for predicting electromagnetic parameters make use of open-source Python libraries and can be obtained from the corresponding authors upon request.

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

## Acknowledgments

Q.D., H.Y., H.L., and L.D. acknowledge the National Natural Science Foundation of China (Project Nos. 51972046 and 52021001). X.G. and Z.C. acknowledge the National Natural Science Foundation of China (Project Nos. 52573145 and 52073302).

## Author contributions

Z.C., T.L., H.L., and L.D. conceived and designed this study. Q.D. fabricated the samples. Q.D., Z.C., T.L., H.L., and L.D. wrote the paper. Q.D., X.G., and H.Y. performed the experiments and analysed the data. Q.D., Z.W., and H.Q. developed a machine learning system for predicting electromagnetic properties. All authors discussed the results and contributed to the scientific interpretation as well as to the writing of the paper. Work by H.Q. was done principally during an internship at A*STAR, IHPC.

## Competing interests

The authors declare no competing interests.
