## [Transparent Peer Review file · Nature Communications]

Adaptive ionic liquid polymer microwave modulation surface with reprogrammable dielectric properties

Corresponding Author: Professor Haipeng Lu

Version 0:

Reviewer comments:

Reviewer #1

(Remarks to the Author)

This manuscript proposes an adaptive ion liquid polymer surface with reprogrammable dielectric properties for microwave modulation. A new electromagnetic modulation strategy is proposed by driving the hydrogen bonds between the ionic liquid and the polymer, utilizing molecular-level orientation motion and changes in ionic mobility. Additionally, the integration of machine learning techniques enables reprogrammable control of dielectric properties, demonstrating excellent microwave modulation capabilities. The paper is highly innovative, with thorough arguments, and the research findings provide valuable insights into the field of dynamic electromagnetic control materials. It offers significant guidance for the development of next-generation smart electromagnetic communication technologies. With appropriate modifications, this paper could be accepted for publication in Nature Communications. Below are some specific suggestions for revision:

1. The programmable dielectric properties are the core method of control in this paper. It is recommended to include relevant dielectric property control results in the introduction (lines 92-96) rather than just focusing on the microwave modulation results. This will help readers gain a more comprehensive understanding of the core content of the paper.
2. The authors mention that the material exhibits good repeatability. Although one set of electromagnetic data after 1000 cycles of temperature cycling is provided, it is recommended to select a representative sample and provide multiple cycles of electromagnetic test data. This would better demonstrate the material's repeatability and enhance the reliability and persuasiveness of the experimental results.
3. In the mechanistic diagram in Figure 2j, it is necessary to label the high-temperature and low-temperature states to help readers clearly understand the changes at different temperatures.
4. The small plot in Figure 3d shows the attenuation coefficient, but the discussion on its impact on electromagnetic performance is insufficient in the main text. It is recommended to add a more detailed explanation to help readers understand the relationship between the attenuation coefficient and electromagnetic performance.
5. The sample name in the lower part of Figure 4e should be "IL-P-6"? Please check the entire manuscript to ensure accuracy of sample names.
6. There are formatting issues with the references in lines 284 and 294. The authors should carefully check and correct these formatting errors to ensure the accuracy of the references.
7. The units of aircraft reported in existing literature are mostly in dBsm. It is recommended to convert the RCS units in Supplementary Fig. 21 to dBsm for easier comparison.

Reviewer #2

(Remarks to the Author)

The authors have developed Ionic Liquid-polymer system with different concentrations of ionic liquid as a microwave absorber. The microwave absorption property is studied as a function of temperature to prove the adaptability and programmable capacity. At the end, pressure dependent property is also demonstrated (I think). The paper is not written in a clear fashion giving confusing statements at several places. In my view, the paper is not suitable for publication in the present form.

The following are a few of my questions related to the manuscript.

1. Abstract does not clearly indicate the total nature of work. Algorithms are useful to predict the dielectric properties but not

ensure programmable nature. Nowhere, the effect of temperature or pressure is indicated in the abstract.

2. The impedance equation mentioned in the supplementary has a typographical mistake. While the absorption relation is fine, why there is a need to indicate the magnetic permeability part since the system is purely dielectric in nature. Infact, the manuscript indicates the prediction of magnetic permeability but nowhere it is reported in the manuscript (except as very low loss tangent in Fig. 3).

3. The microwave dielectric properties are measured using a coaxial transmission line technique (assumed from the information on the toroidal sample). If so, how did they obtained RL from the experiment?

4. The sentence "The permittivity ($\epsilon_r = \epsilon' - j\epsilon''$) was measured to evaluate the dielectric properties in the microwave frequency range (2 to 18 GHz)." is a bit confusing and may be rephrased properly. The same paragraph cites the dielectric data variation with temperature stimulation in a different frequency range also.

5. The dielectric parameters do not match when compared with fig. 7 and 8 (supplementary) for the two frequency ranges. While VNA shows high value of dielectric permittivity at <1 GHz, the low frequency data shows a very low value. It would be better if the measurements or simulations are limited to the frequency range of application.

6. Since both simulation and experiment are performed, the authors should clearly state what are the graphs are from experiment and what graphs are plotted from simulation. Only for the reflection loss measurements comparison is provided.

7. It is a bit confusing when the authors mention as "the ability of the IL-P surface to manipulate the electromagnetic waves". The sample, being a homogeneous in nature (I assume), why only the surface is important when the thickness of the sample is approximately 2 mm?

8. The application of the material is for the modifying the microwave absorption properties with the external stimuli. This is going to be a slow process and may not be of use in any practical applications such as Radar, active stealth technology etc.

Reviewer #3

(Remarks to the Author)

The authors highlight the importance of reconfigurable metasurfaces and propose an ionic liquid (IL)–based reconfigurable metasurface operating at microwave frequencies as the key originality of this work. The presented theoretical analysis, simulation results, fabrication process, and measurement data are of high quality and clearly described. However, the reviewer finds that the novelty claimed by utilizing IL for metasurface reconfigurability is not sufficiently justified. The detailed reasoning is as follows:

1. The authors argue that the first implementation of IL for tunable metasurfaces at microwave frequencies constitutes the main novelty of this manuscript. However, IL-based reconfigurable or absorptive metasurfaces have already been well reported in prior literature. The novelty of this work compared with existing contributions is therefore unclear. Representative examples include:

Fulong Yang, Zhinan Shi, Zhitao Guo, Lijun Gong, Jinyan Wei, Teng Wang, Zhiwen Wang, High sensitivity metasurface sensor for estimating the complex permittivity of ionic liquids, *Sensors and Actuators A: Physical*, Volume 377, 2024, 115737, <https://doi.org/10.1016/j.sna.2024.115737>.

Zirui Yu, Yuyang Wang, Jiajun Fan, Xiaoya Zhang, Yongji Guan, Imidazole ionic liquid based ultra-broadband metamaterial absorbers with a concave–convex structure, *Journal of Physics D: Applied Physics*, 10.1088/1361-6463/adde6c, 58, 24, (245501), (2025).

Gong J, Yang F, Shao Q, He X, Zhang X, Liu S and Deng Y 2017 Microwave absorption performance of methylimidazolium ionic liquids: towards novel ultra-wideband metamaterial absorbers *RSC Adv.* 7 41980–8

Jie Luo, Xiang Fang, Xiao Liu, Zhuang Wu, Yanan Zeng, Yuntao Yang, Wenxing Zou, Shi Qiao, Qian Xue, Jiayi Xiong, Hongbin Fei, Yanhong Zou, Functional Multispectral Camouflage Strategy Based on Flexible Transparent Metamaterial Compatible with Radiative Cooling, *Laser & Photonics Reviews*, 10.1002/lpor.202401905, 19, 12, (2025).

2. Although the authors demonstrate that IL can effectively tune electromagnetic properties, liquid crystals (LCs) are also widely used for tunable metasurface components. A thorough comparison between the proposed IL-based approach and well-known LC-based implementations is necessary to clarify the relative advantages.

3. Beyond IL component, many active components have been investigated for tunable metasurfaces (e.g., LCs, PIN, varactors diodes, phase-change materials, mechanical deformation,). The authors should explicitly state the unique benefits of IL-based reconfigurability in comparison with these alternatives.

4. The manuscript demonstrates the proposed metasurface for stealth applications. However, in state-of-the-art stealth technology, additional factors such as broadband and continuously frequency-reconfigurable performance (not just discrete switching), operation under oblique incidence for bi-static RADAR detection, switching speed, robustness under harsh environments (e.g., extremely low temperatures), and repeatability are all critical considerations. These aspects should be further addressed to strengthen the practical relevance of the proposed work.

Version 1:

Reviewer comments:

Reviewer #1

(Remarks to the Author)

This manuscript has been carefully revised according to referee's comments. I would like to recommend the acceptance of it

for publication.

Reviewer #2

(Remarks to the Author)

The paper may be accepted if the work is limited to the modification of dielectric parameters and the related analysis using the chemistry of materials. The application part must be pruned since it deviates from the main aspect of the paper. This is due to the following:

1. The authors in their reply to the authors query mention that the main focus is not microwave absorption.
2. The application emphasised is the switchable microwave absorbers which is good for slow speed applications and not definitely for radar applications.
3. While the material does not have magnetic part, the evaluation of magnetic parameters at microwave frequencies are obtained without using the proper model meant only for dielectric materials.
4. The equation for RL (S8) still has a problem since it uses $(Z_{in}-1)/(Z_{in}+1)$ while Z_{in} is not a normalized quantity.

Reviewer #3

(Remarks to the Author)

The manuscript reports a thermally tunable microwave surface based on ionic liquid–polymer (IL–P) composites, showing potential for adaptive electromagnetic applications.

However, two major conceptual issues remain: (1) the authors' overly narrow interpretation of what constitutes a reconfigurable metasurface, and (2) the lack of a functional comparison with well-established liquid crystal (LC)-based tunable metasurfaces.

These conceptual inconsistencies obscure the true positioning and novelty of the work.

Comment 1:

The authors argue that their work should not be categorized as a reconfigurable metasurface.

This argument is unconvincing and inconsistent with the current understanding of reconfigurable metasurfaces.

By widely accepted definitions, a reconfigurable metasurface refers to any electromagnetic surface whose scattering or absorption properties can be dynamically altered through external stimuli, whether the underlying mechanism is:

- mechanical deformation,
- electrical bias or carrier tuning, or
- material-level modulation, such as temperature-driven ionic or molecular reorientation.

In the present study, the authors clearly describe a temperature-induced modulation of dielectric properties through hydrogen-bond reconstruction and ionic motion within the IL–P matrix.

This results in measurable shifts of reflection loss and absorption bandwidth—i.e., a dynamic reconfiguration of electromagnetic response.

Such behavior squarely fits the definition of a reconfigurable metasurface, even in the absence of mechanical actuation.

Moreover, the manuscript itself repeatedly employs terminology that explicitly conveys reconfigurability—namely:

- "Adaptive ionic liquid polymer", "microwave modulation", "Reprogrammable dielectric properties"
- "Reconstruction of hydrogen bonds", "Controllable modulation of dielectric properties at microwave frequencies", "Switchable microwave absorbing", "Tunable effective absorption"

These expressions inherently describe stimuli-responsive electromagnetic platforms whose properties can be tuned or reprogrammed.

It is therefore inconsistent for the authors to employ this language while simultaneously claiming that their device is not a reconfigurable metasurface.

This distinction appears semantic rather than technical, and it undermines the conceptual clarity of the work.

Comment 2:

The authors' response regarding liquid crystal (LC)-based tunable metasurfaces does not sufficiently address the reviewer's concern.

The question was not about the chemical phase (liquid vs. solid), but about the functional analogy between IL–P and LC systems.

Both rely on orientation-dependent polarization and external-stimulus-driven molecular or ionic reordering to achieve dielectric and electromagnetic tunability.

Thus, their operating principles are physically comparable.

To substantiate the claimed novelty of the IL–P approach, the authors should provide a quantitative or conceptual comparison with LC-based metasurfaces, such as:

- the range and reversibility of dielectric tunability ($\Delta\epsilon'$, $\Delta\epsilon''$),
- response time and operational stability,
- frequency range (microwave vs. optical), and

- fabrication scalability or robustness.

Without such analysis, the distinction between IL–P and LC systems remains superficial, and the claimed advantage of the IL–P-based approach is not convincingly demonstrated.

Version 2:

Reviewer comments:

Reviewer #2

(Remarks to the Author)

The authors have adequately replied to the queries and modified the manuscript accordingly. The manuscript may be accepted.

Reviewer #3

(Remarks to the Author)

The authors have provided appropriate responses to the reviewer's questions, and I believe this manuscript is suitable for publication in Nature Communications.

Reviewer #1:

This manuscript proposes an adaptive ion liquid polymer surface with reprogrammable dielectric properties for microwave modulation. A new electromagnetic modulation strategy is proposed by driving the hydrogen bonds between the ionic liquid and the polymer, utilizing molecular-level orientation motion and changes in ionic mobility. Additionally, the integration of machine learning techniques enables reprogrammable control of dielectric properties, demonstrating excellent microwave modulation capabilities. The paper is highly innovative, with thorough arguments, and the research findings provide valuable insights into the field of dynamic electromagnetic control materials. It offers significant guidance for the development of next-generation smart electromagnetic communication technologies. With appropriate modifications, this paper could be accepted for publication in Nature Communications. Below are some specific suggestions for revision:

1. The programmable dielectric properties are the core method of control in this paper. It is recommended to include relevant dielectric property control results in the introduction (lines 92-96) rather than just focusing on the microwave modulation results. This will help readers gain a more comprehensive understanding of the core content of the paper.

Thank you for the reviewer's suggestion. We have added the relevant dielectric property control results in the introduction section (lines 92-96):

In the "off-to-on" mode, ϵ' increases from 4.79 to 6.43, and ϵ'' increases from 1.06 to 2.78 at 14.76 GHz. The tunable effective absorption bandwidth (ΔEAB) reaches up to 5.69 GHz, with reflection loss (RL) values ranging from -6.04 dB to -46.21 dB. In contrast, in the "on-to-off" mode, ϵ' ranges from 7.73 to 11.20, and ϵ'' ranges from 3.14 to 6.98 at 13.63 GHz, exhibiting a ΔEAB of up to 5.34 GHz and RL values ranging from -50.48 dB to -6.47 dB.

2. The authors mention that the material exhibits good repeatability. Although one set of electromagnetic data after 1000 cycles of temperature cycling is provided, it is recommended to select a representative sample and provide multiple cycles of electromagnetic test data. This would better demonstrate the material's repeatability and enhance the reliability and persuasiveness of the experimental results.

Thank you for the reviewer's suggestion. We have added the explanation regarding the repeatability in lines 323-325 of the main text:

In addition, the repeatability of the electromagnetic switching in active microwave absorbing surfaces is crucial for practical applications. IL-P-2 was selected as a representative sample and demonstrated extremely stable electromagnetic control characteristics through high-low temperature cycling tests.

In addition, we have added Supplementary Fig. 19 in the supplementary materials to further illustrate the repeatability of the material:

Supplementary Fig. 19 | The electromagnetic parameter cycling repeatability test for IL-P-2. a Real part of permittivity. b Imaginary part of permittivity. c Real part of the permeability. d Imaginary part of the permeability.

- In the mechanistic diagram in Figure 2j, it is necessary to label the high-temperature and low-temperature states to help readers clearly understand the changes at different temperatures.

Thank you for the reviewer's suggestion. We have added the high-temperature and low-temperature states in Fig. 2j:

Fig. 2 | The effect of $[\text{EtA}^+][\text{NO}_3^-]$ mobility in the PHEA on electromagnetic properties. **a** In situ FTIR of IL-P-2 under thermal stimulation. **b** and **c** VT-FTIR synchronous and asynchronous correlation spectroscopy. **d** T_1 and T_2 spectra of IL-P-2 in low-field NMR under different temperature stimuli. **e**, **f** 2D low-field ^1H NMR spectra at 30°C and 130°C . **(G)** ϵ' of IL-P-0 and IL-P-2. **h** Dielectric loss fitting results for IL-P-0, red: conductivity loss, blue: polarization loss. **i** Dielectric loss fitting results for IL-P-2, red: conductivity loss induced by IL, blue: polarization loss induced by PHEA. **j** Mechanism of temperature stimulus on the ϵ'' of IL-P

4. The small plot in Figure 3d shows the attenuation coefficient, but the discussion on its impact on electromagnetic performance is insufficient in the main text. It is recommended to add a more detailed explanation to help readers understand the relationship between the attenuation coefficient and electromagnetic performance.

Thank you for the reviewer's helpful suggestion. We noticed that we lacked a discussion on the attenuation coefficient in the main text, and we have made the following additions (lines 297-301):

Therefore, based on the trend of these electromagnetic parameters changing with temperature, it can be concluded that the attenuation coefficient is positively correlated with both temperature and the IL concentration in the polymer (Fig. 3d). This indicates that higher temperatures lead to greater attenuation of microwaves during propagation through the IL-P.

5. The sample name in the lower part of Figure 4e should be "IL-P-6"? Please check the entire manuscript to ensure accuracy of sample names.

Thank you for the reviewer's careful review. We have corrected this printing error and have thoroughly checked the entire manuscript.

Fig. 4 | Large-area switchable microwave modulation surfaces. a

Demonstration of large-area flexible IL-P. **b** Far-field scanning microwave reflection images of a single 2mm-thick IL-P-2 surface at different temperatures, with an area of 10×10 cm. **c, d** Pixelated microwave absorbing surface formed by scanned microwave reflection of a 3×3 array of nine units with different temperature configurations. **e-g** Single-station RCS of a 2mm-thick switchable radar absorbing surface, with an area of 10×10 cm.

6. There are formatting issues with the references in lines 284 and 294. The authors should carefully check and correct these formatting errors to ensure the accuracy of the references.

Thank you for the reviewer's verification. We have corrected this error and have thoroughly checked the entire manuscript.

7. The units of aircraft reported in existing literature are mostly in dBsm. It is recommended to convert the RCS units in Supplementary Fig. 21 to dBsm for easier comparison.

Thank you for the reviewer's suggestion. We have converted the unit of RCS to dBsm:

Supplementary Fig. 22 | Switchable RCS on the aircraft surface. a-c Simulated RCS (dBsm) results of 2 mm-thick IL-P-2 at different cross-sections at 30°C and 130°C. d-f Simulated RCS (dBsm) results of 2 mm-thick IL-P-6 at different cross-sections at 30°C and 130°C.

Reviewer #2:

The authors have developed Ionic Liquid-polymer system with different concentrations of ionic liquid as a microwave absorber. The microwave absorption property is studied as a function of temperature to prove the adaptability and programmable capacity. At the end, pressure dependent property is also demonstrated (I think). The paper is not written in a clear fashion giving confusing statements at several places. In my view, the paper is not suitable for publication in the present form. The following are a few of my questions related to the manuscript.

1. Abstract does not clearly indicate the total nature of work. Algorithms are useful to predict the dielectric properties but not ensure programmable nature. Nowhere, the effect of temperature or pressure is indicated in the abstract.

We would like to thank the reviewer for their insightful comments.

First, we have reorganized and revised the abstract to clearly explain the role of machine learning in the study, in order to avoid any unnecessary misunderstandings:

Adaptive microwave surfaces have the capability to dynamically adjust their electromagnetic transmission to meet specific needs, offering significant potential for efficient integration and flexible use in reconfigurable communication systems. In this work, we utilize temperature induced break and reconstruction of hydrogen bonds to drive the orientational motion and charge mobility of the ionic liquid [EtA⁺][NO₃⁻] in the PHEA, resulting controllable modulation of dielectric properties at microwave frequencies. Building on this mechanism, we applied machine learning algorithms to establish correlations between temperature, ionic liquid concentration, and dielectric constant, enabling the design of a reprogrammable dielectric microwave modulation surface. For example, the 2 mm-thick switchable microwave absorbing surfaces fabricated here can operate in two distinct modes during the temperature transition from low to high, namely “off-to-on” and “on-to-off”. The corresponding tunable effective absorption bandwidths and reflection loss values reach 5.69 GHz, -6.04 dB to -46.21 dB, and 5.34 GHz, -50.48 dB to -6.47 dB, respectively. Using the developed active surface, we also demonstrate various 3D-printed device architectures, including pixelated surfaces, metasurfaces, and self-sensing functionalities, which provide valuable guidance for the development of next-generation intelligent electromagnetic devices.

We also completely agree that a predictive algorithm by itself is not equivalent to programmability. In our work, the core of "reprogrammability" lies in the establishment of a "target-predict-drive" active control loop.

Goal setting: We first set a desired electromagnetic response goal (e.g., achieving a -XX dB reflection loss at XX GHz frequency).

Algorithm prediction: Next, our trained machine learning model acts as the "brain," which, based on this goal, predicts the optimal material composition (ionic liquid concentration) and external stimulus parameters (temperature) required to achieve the goal.

Active driving: Finally, we actively "program" the dielectric properties to the desired state through precise control of the temperature control device, thereby achieving the intended electromagnetic modulation.

Therefore, the role of the machine learning algorithm here is not just "prediction," but rather serves as the intelligent control core of this programmable system. It establishes a quantitative mapping between the "stimulus parameters" and the "electromagnetic response," enabling us to reverse-engineer and tailor the material's performance as needed. We noticed that this point was not well explained in the manuscript, and we have made the necessary additions to clarify it (lines 308-310):

Using this model and data foundation, we established a "target-prediction-drive" active control loop, achieving programmable design of microwave absorption performance, and successfully constructed two switchable state microwave absorbing surfaces.

We would like to thank the reviewer for their reminder. However, the effect of temperature is already mentioned at the beginning of the second sentence of the abstract.

In this work, we utilize temperature induced break and reconstruction of hydrogen bonds to drive the orientational motion and charge mobility of the ionic liquid [EtA⁺][NO₃⁻] in the PHEA, resulting controllable modulation of dielectric properties at microwave frequencies.

Additionally, we have included information about the temperature in the description of the abstract:

For example, the 2 mm-thick switchable microwave absorbing surfaces fabricated here can operate in two distinct modes during the temperature transition from low to high, namely "off-to-on" and "on-to-off".

However, we believe that the impact of pressure does not need to be mentioned in the abstract, as the sensing performance is inherently influenced by pressure, and it is merely an additional property of the material, not the primary focus of the study.

2. The impedance equation mentioned in the supplementary has a typographical mistake. While the absorption relation is fine, why there is a need to indicate the magnetic permeability part since the system is purely dielectric in nature. Infact, the manuscript indicates the prediction of magnetic permeability but nowhere it is reported in the manuscript (except as very low loss tangent in Fig. 3).

Thank you for your insightful suggestion. The issue you raised is very professional and to the point. It highlights a potential area of confusion for readers in the manuscript. Let us clarify this matter in detail.

Firstly, we have corrected the printing error in the impedance equation:

$$Z_{in} = Z_0 \sqrt{\frac{\mu_r}{\epsilon_r}} \tanh \left[j \frac{2\pi f d}{c} \sqrt{\mu_r \epsilon_r} \right]$$

From the above equation, it can be seen that the performance of electromagnetic wave control materials is intrinsically related to the changes in magnetic permeability. Only when the magnetic permeability remains constant can changes in dielectric properties be both a necessary and sufficient condition for controlling electromagnetic waves. In Fig. 3d, we report the magnetic loss tangent, and in Figs. S12 and S13, we present the real and imaginary parts of the magnetic permeability. Furthermore, in lines 294-297 of the main text, we explicitly state:

“Interestingly, due to the stability of the magnetic permeability (Fig. 3d), the transmission

behavior almost entirely depends on the modulation of the aforementioned dielectric properties, making the IL concentration a key factor for the electromagnetic wave control performance.”

3. The microwave dielectric properties are measured using a coaxial transmission line technique (assumed from the information on the toroidal sample). If so, how did they obtained RL from the experiment?

The RL (reflection loss) value is derived from the electromagnetic parameters measured using coaxial line techniques, and it is obtained through the dielectric constant and magnetic permeability constants. This can be referenced in the following literature:

[1] Z. Cheng et al., Intelligent Off/On Switchable Microwave Absorption Performance of Reduced Graphene Oxide/VO₂ Composite Aerogel. *Advanced Functional Materials* 32, 2205160 (2022).

[2] X. Zhang et al., Metal-organic frameworks with fine-tuned interlayer spacing for microwave absorption. *Science Advances* 10, ead16498 (2024).

This part may have caused some confusion, so we have provided clarification in the figure caption of Fig. 3:

Fig3 f The matching degree of the corresponding wave impedance with free space impedance ($Z_0 = 377$), Z_{in}/Z_0 , and the normal direction RL of the 2 mm-thick IL-P-2 and IL-P-6 surfaces from 0.5 to 18 GHz under different temperature stimuli are calculated based on the measured electromagnetic parameters.

The formula for **RL** is given by:

$$Z_{in} = Z_0 \sqrt{\frac{\mu_r}{\epsilon_r}} \tanh \left[j \frac{2\pi f d}{c} \sqrt{\mu_r \epsilon_r} \right]$$

$$\alpha = \frac{\sqrt{2}\pi f}{c} \times \sqrt{(\mu''\epsilon'' - \mu'\epsilon') + \sqrt{(\mu''\epsilon'' - \mu'\epsilon')^2 + (\mu''\epsilon' + \mu'\epsilon'')^2}}$$

$$RL(dB) = 20 \log_{10} \frac{|Z_{in} - 1|}{|Z_{in} + 1|}$$

4. The sentence "The permittivity ($\epsilon_r = \epsilon' - j\epsilon''$) was measured to evaluate the dielectric properties in the microwave frequency range (2 to 18 GHz)." is a bit confusing and may be rephrased properly. The same paragraph cites the dielectric data variation with temperature stimulation in a different frequency range also.

Thank you for the reviewer’s suggestion. We have made the following modifications (lines 216-218):

“The dielectric constant ($\epsilon_r = \epsilon' - j\epsilon''$) of the material under different temperature stimuli was measured to evaluate its dielectric response in the 2 - 18 GHz microwave frequency range using a vector network analyzer. The dielectric relaxation behavior of the polymer in IL-P was assessed using an impedance analyzer in the frequency range of 10^7 to 5×10^8 Hz.”

As the reviewer mentioned, we measured the dielectric constant at low frequencies because it provides a more comprehensive mechanistic understanding of the source of the dielectric constant at high frequencies (microwave frequency range). This is because the ionic liquid–

polymer (IL-P) system is a composite material, and the polymer component also exhibits dielectric changes due to dipole orientation under temperature stimuli. At low frequencies, we can observe the variation in the relaxation peak associated with the dipole orientation motion of the polymer system, from which we can determine the dipole orientation relaxation time (lines 230-256). Based on the relaxation time, we can then apply the H-N equation:

$$\varepsilon = \varepsilon' - i\varepsilon'' = -i \left(\frac{\sigma_0}{\varepsilon_0 \omega} \right)^N + \sum_{k=1}^2 \left[\frac{\Delta\varepsilon_k}{(1 + (i\omega\tau_k)^{\alpha_k})^{\beta_k}} + \varepsilon_{\infty,k} \right]$$

This allows us to determine the contribution ratio of the dipolar relaxation loss (ε''_p) from the polymer and the conduction loss (ε''_c) from the ionic liquid to the total dielectric loss (ε''), which in turn leads to the data presented in Figures 2h and 2i. The contributions of the dielectric properties under these temperature stimuli provide mechanistic evidence, which is discussed in detail in combination with infrared and low-field nuclear magnetic resonance molecular dynamics analysis. This section (lines 113-256 of the main text, Figs. 1 and 3) highlights the mechanism through which dipolar ions, at the molecular level, in the polymer influence the material's electromagnetic parameters. It is precisely by clarifying this control mechanism that we can confirm the dielectric modulation trend is stable and systematic, thereby providing a rich theoretical foundation for the subsequent programmable design of dielectric properties through machine learning.

5. The dielectric parameters do not match when compared with fig. 7 and 8 (supplementary) for the two frequency ranges. While VNA shows high value of dielectric permittivity at <1 GHz, the low frequency data shows a very low value. It would be better if the measurements or simulations are limited to the frequency range of application.

We would like to thank the reviewer for their thorough review. The confusion arose due to our failure to specify the testing conditions. First, the low-frequency measurements were conducted using an impedance analyzer, while the high-frequency measurements were performed using a vector network analyzer. We have now added this clarification in the main text (lines 513-515):

Electromagnetic parameters were measured using the Agilent E4991 impedance analyzer over the frequency range of 10^7 to 5×10^8 Hz, with the material tested in the form of thin discs with a thickness of approximately 1 mm and a diameter of 7 mm.

As the frequency decreases, conductive loss becomes more pronounced; therefore, a high apparent dielectric constant at low frequencies (below 1 GHz) is expected, consistent with the following relations:

$$\varepsilon''_c = \frac{\sigma}{\omega \varepsilon_0}$$

$$\omega = 2\pi f$$

Additionally, we selected some data from Fig. 7 and 8 (supplementary), and when plotted on the same graph, it can be observed that their differences are not significant.

Additionally, as explained in our previous response, the low-frequency data were measured solely to observe the relaxation peaks of the polymer, which allowed us to determine the relaxation time for dipole orientation. Therefore, the deviation in the low-frequency values does not affect the fitting results of the H-N equation.

In our subsequent studies, the machine learning predictions and simulation results were all conducted within the application frequency range of 0.5-18 GHz, as shown in Figs. 3 and S15-17.

6. Since both simulation and experiment are performed, the authors should clearly state what are the graphs are from experiment and what graphs are plotted from simulation. Only for the reflection loss measurements comparison is provided.

Thank you for the reviewer's suggestions. The questions you raised are very professional and pertinent. They pointed out a potential area of confusion in the manuscript for the readers. This was an oversight on our part, and we have now provided a clear explanation in the figure caption and thoroughly checked the manuscript for similar errors. Additionally, in the inset of the figure, we provided a simulation of the dielectric values (dashed line), but it was not clearly visible. We have now lightened the color of the solid line to make the dashed line more distinguishable.

Fig. 3 | Reprogrammable microwave absorbing surfaces composed of IL-P with different IL concentrations. a-c ϵ' , ϵ'' , and $\tan \delta_\epsilon$ of IL-P at 10 GHz under different temperature stimuli. **d** $\tan \delta_\mu$ of IL-P at 10 GHz under different temperature stimuli, with the attenuation coefficient shown in the upper right corner. **e** Modeling and prediction of electromagnetic parameters and RL under multivariable coupling conditions based on machine learning algorithms, where ϵ'_n , ϵ''_n , μ'_n , μ''_n , represent the predicted values (dashed lines: predicted values, solid lines: actual values). **f** The matching degree of the corresponding wave impedance with free space impedance ($Z_0 = 377$), Z_{in}/Z_0 , and the normal direction RL of the 2 mm-thick microwave modulation surfaces composed of IL-P-2 and IL-P-6 in the 0.5 to 18 GHz range under different temperature stimuli are calculated based on the measured electromagnetic parameters. **g** Δ EAB of IL-P under different temperature stimuli. **h** RL of the switchable microwave absorbing surface for microwaves at incident angles, ranging from -80° to 80° .

- It is a bit confusing when the authors mention as "the ability of the IL-P surface to manipulate the electromagnetic waves". The sample, being a homogeneous in nature (I assume), why only the surface is important when the thickness of the sample is

approximately 2 mm?

We thank the reviewer for their question. By "surface," we are referring to the switchable microwave-absorbing surface, which is composed of a 2 mm thick IL-P material, rather than the "surface" of a specific object. We have already emphasized this in the main text, for example, in lines 90-92:

For example, using this method, 2 mm-thick switchable microwave absorbing surfaces composed of IL-P with two different IL concentrations were designed, which exhibited distinct switching modes during the temperature increase process.

This terminology is commonly found in the literature on tunable microwave properties, for example:

Li, W., Xu, M., Xu, H. X., Wang, X. & Huang, W. Metamaterial Absorbers: From Tunable Surface to Structural Transformation. *Adv. Mater.* 34, 2202509 (2022).

We noticed that the term "IL-P surface" could indeed cause confusion. We have corrected this expression, for example, by removing "IL-P-based active microwave absorbing surfaces" or describing it as "microwave modulation surfaces composed of IL-P."

8. The application of the material is for the modifying the microwave absorption properties with the external stimuli. This is going to be a slow process and may not be of use in any practical applications such as Radar, active stealth technology etc.

We appreciate the reviewer's suggestion. We believe that the main innovation of this work lies in the impact of ionic liquids (at the molecular level) on high-frequency (microwave) dielectric properties, as well as the programmable design of dielectric changes through machine learning, based on temperature stimuli and ionic liquid concentration. The innovation in the mechanism and proof of this programmable dielectric performance is of greater significance. Microwave absorption control is just one example of dielectric modulation and not the primary focus.

Moreover, the decisive factor for the speed of modulation is the rate of external stimulus control, which depends on the power of the heater. Current studies show that temperature switching can be completed within seconds, as demonstrated in the following references:

[1] F. Xu. et al. Highly stretchable, fast thermal response carbon nanotube composite heater, *COMPOS PART A-APPL S.* 147 (2021) 106471. doi.org/10.1016/j.compositesa.2021.106471.

[2] B. Zhou. et al. Ultrathin, flexible transparent Joule heater with fast response time based on single-walled carbon nanotubes/poly(vinyl alcohol) film, *Compos. Sci. Technol.* 183 (2019) 107796. doi.org/10.1016/j.compscitech.2019.107796.

Furthermore, there are numerous reports in the literature on microwave modulation using temperature as an external stimulus, although their control mechanisms differ from ours:

[1] Cheng, Z. et al. Intelligent Off/On Switchable Microwave Absorption Performance of Reduced Graphene Oxide/VO₂ Composite Aerogel. *Adv. Funct. Mater.* 32, 2205160 (2022).

[2] Hou, Y., Sheng, Z., Fu, C., Kong, J. & Zhang, X. Hygroscopic holey graphene aerogel fibers enable highly efficient moisture capture, heat allocation and microwave absorption. *Nat. Commun.* 13, 1227 (2022).

[3] Lepeshov, S. & Krasnok, A. Tunable phase-change metasurfaces. *Nat. Nanotechnol.* 16, 615-616 (2021).

[4] Wei, H. et al. Tunable VO₂ cavity enables multispectral manipulation from visible to microwave frequencies. *Light-Sci. Appl.* 13, 54 (2024).

[5] Liao, S. Y. et al. Reversible Switching Between Microwave Absorption and EMI Shielding of VO₂ Composite Foam. *Small* 20, 2402841 (2024).

Reviewer #3:

The authors highlight the importance of reconfigurable metasurfaces and propose an ionic liquid (IL) - based reconfigurable metasurface operating at microwave frequencies as the key originality of this work. The presented theoretical analysis, simulation results, fabrication process, and measurement data are of high quality and clearly described. However, the reviewer finds that the novelty claimed by utilizing IL for metasurface reconfigurability is not sufficiently justified. The detailed reasoning is as follows:

1. The authors argue that the first implementation of IL for tunable metasurfaces at microwave frequencies constitutes the main novelty of this manuscript. However, IL-based reconfigurable or absorptive metasurfaces have already been well reported in prior literature. The novelty of this work compared with existing contributions is therefore unclear. Representative examples include:

Fulong Yang, Zhinan Shi, Zhitao Guo, Lijun Gong, Jinyan Wei, Teng Wang, Zhiwen Wang, High sensitivity metasurface sensor for estimating the complex permittivity of ionic liquids, *Sensors and Actuators A: Physical*, Volume 377, 2024, 115737, <https://doi.org/10.1016/j.sna.2024.115737>.

Zirui Yu, Yuyang Wang, Jiajun Fan, Xiaoya Zhang, Yongji Guan, Imidazole ionic liquid based ultra-broadband metamaterial absorbers with a concave - convex structure, *Journal of Physics D: Applied Physics*, 10.1088/1361-6463/adde6c, 58, 24, (245501), (2025).

Gong J, Yang F, Shao Q, He X, Zhang X, Liu S and Deng Y 2017 Microwave absorption performance of methylimidazolium ionic liquids: towards novel ultra-wideband metamaterial absorbers *RSC Adv.* 7 41980 - 8

Jie Luo, Xiang Fang, Xiao Liu, Zhuang Wu, Yanan Zeng, Yuntao Yang, Wenxing Zou, Shi Qiao, Qian Xue, Jiayi Xiong, Hongbin Fei, Yanhong Zou, Functional Multispectral Camouflage Strategy Based on Flexible Transparent Metamaterial Compatible with Radiative Cooling, *Laser & Photonics Reviews*, 10.1002/lpor.202401905, 19, 12, (2025).

We appreciate the reviewer's suggestion. First of all, we do not consider "IL-based reconfigurable or absorptive metasurfaces" as the highlight of this paper. In fact, we do not even mention the concept or term "reconfigurable metasurfaces" anywhere in the manuscript. Our primary control mechanism is based on the orientation movement and mobility of ionic liquids, along with changes in ionic conductivity in the polymer matrix, which govern the intrinsic dielectric properties of the material. Figs. 1-3, which make up a substantial part of the manuscript, focus on explaining this control mechanism, which is the key to the innovation of our paper. In contrast, in the first paragraph, we extensively reference literature to highlight the limitations of "reconfigurable metasurfaces" in microwave control mechanisms. Therefore, we emphasize the need to explore functional electromagnetic materials that induce changes in the intrinsic electromagnetic properties of the material:

“For instance, arrayed active circuit elements assembled with passive metallic structures have been investigated^{1,3-5,12}. Since external stimuli primarily affect the micron-scale circuit components within the array, the fabrication of large-area surfaces and the microwave control effects are limited. Other studies have typically focused on tuning the permittivity of materials, for example...”

[1] Li, F. et al. Flexible intelligent microwave metasurface with shape-guided adaptive programming. *Nat. Commun.* **16**, 3161 (2025).

[3] Qian, C. et al. Deep-learning-enabled self-adaptive microwave cloak without human intervention. *Nat. Photonics* **14**, 383-390 (2020).

[4] Lim, D. D. et al. A tunable metamaterial microwave absorber inspired by chameleon's color-changing mechanism. *Sci. Adv.* **11**, eads3499 (2025).

[5] Li, W., Xu, M., Xu, H. X., Wang, X. & Huang, W. Metamaterial Absorbers: From Tunable Surface to Structural Transformation. *Adv. Mater.* **34**, 2202509 (2022).

[12] Saifullah, Y., He, Y., Boag, A., Yang, G. M. & Xu, F. Recent Progress in Reconfigurable and Intelligent Metasurfaces: A Comprehensive Review of Tuning Mechanisms, Hardware Designs, and Applications. *Adv. Sci.* **9**, 2203747 (2022).

The reviewer may have misunderstood the exploratory nature of our work in Fig. 5. The IL-P dielectric tunable material was structured as a metasurface, but this was done solely to show that introducing a metasurface could shift the microwave "switching" control band to lower frequencies. We 3D printed the material into a metasurface structure, but it is not a "reconfigurable metasurface" as mentioned by the reviewer, since each metasurface unit does not possess any "reconfigurability".

We noticed that the four references cited by the reviewer primarily focus on the mechanism of controlling the type of ionic liquid to influence microwave absorption. This is not directly related or comparable to our approach of dynamically tuning the microwave absorption properties of ionic liquid – polymer (IL-P) composites through external stimuli. It is important to note that although ionic liquids and ionic liquid – polymers have similar names, their chemical structures, physical properties, and application domains are significantly different. The most prominent distinction is that ionic liquids are in a liquid state, while ionic liquid – polymers are in a solid state.

Fulong Yang, Zhinan Shi, Zhitao Guo, Lijun Gong, Jinyan Wei, Teng Wang, Zhiwen Wang, High sensitivity metasurface sensor for estimating the complex permittivity of ionic liquids, *Sensors and Actuators A: Physical*, Volume 377, 2024, 115737, <https://doi.org/10.1016/j.sna.2024.115737>. This paper studies the microwave metasurface sensors of four different types of ionic liquids: [EMIm][BF₄], [EMIm][OTf], [BMIm][SCN], [BMIm][OTf], [HMIm][NTF₂], and [OMIm][BF₄]. However, it does not demonstrate any effect of external dynamic stimuli on microwave modulation.

Zirui Yu, Yuyang Wang, Jiajun Fan, Xiaoya Zhang, Yongji Guan, Imidazole ionic liquid based ultra-broadband metamaterial absorbers with a concave–convex structure, *Journal of Physics D: Applied Physics*, 10.1088/1361-6463/adde6c, 58, 24, (245501), (2025). This paper investigates a novel metasurface absorber based on ionic liquids with a concave-convex structure. It achieves excellent wave absorption performance across ultra-wideband frequencies, with polarization insensitivity, wide-angle incidence, and temperature stability, through a combination of material selection (ionic liquid), structural innovation (concave-

convex design) of the container, and packaging optimization. There is also a distinction between ionic liquids and ionic liquid polymers in terms of material composition. Moreover, it does not exhibit any microwave control characteristics.

Gong J, Yang F, Shao Q, He X, Zhang X, Liu S and Deng Y 2017 Microwave absorption performance of methylimidazolium ionic liquids: towards novel ultra-wideband metamaterial absorbers RSC Adv. 7 41980–8. This article is the same as the first one cited by the reviewer, which studies the microwave absorption properties of metasurfaces formed by ionic liquids such as [EMIm][Ac], [EMIm][BF₄], [EMIm][N(CN)₂], and [EMIm][NTf₂] in circular containers. It similarly does not demonstrate any microwave modulation under external dynamic stimuli.

Jie Luo, Xiang Fang, Xiao Liu, Zhuang Wu, Yanan Zeng, Yuntao Yang, Wenxing Zou, Shi Qiao, Qian Xue, Jiayi Xiong, Hongbin Fei, Yanhong Zou, Functional Multispectral Camouflage Strategy Based on Flexible Transparent Metamaterial Compatible with Radiative Cooling, Laser & Photonics Reviews, 10.1002/lpor.202401905, 19, 12, (2025). This article primarily discusses multifunctional and multispectral compatible camouflage technology, which is not directly related to our research on achieving microwave modulation under external dynamic stimuli.

I believe the following figure can clearly illustrate the differences between these references and our research:

Fig. (a) Literature provided by Reviewer. (b) Metasurface structure constructed using ionic liquid polymer in this work.

In conclusion, we kindly ask the reviewer to reconsider the innovation of our paper.

2. Although the authors demonstrate that IL can effectively tune electromagnetic properties, liquid crystals (LCs) are also widely used for tunable metasurface components. A thorough comparison between the proposed IL-based approach and well-known LC-based implementations is necessary to clarify the relative advantages.

We appreciate the reviewer's question. First, we would like to reiterate that the material we

used is not ionic liquid (IL). The main material we are regulating is ionic liquid – polymer (IL-P). Our method involves using external stimuli to break the hydrogen bonds between the polymer and ionic liquid in IL-P, which drives the orientation movement of the ionic liquid and changes its ion mobility. In this context, the ionic liquid is confined within the polymer matrix. Our study focuses on understanding how the ionic liquid within the confined polymer matrix alters the dielectric properties and how machine learning can establish the relationship between temperature, ionic liquid concentration, and the resulting dielectric changes.

Regarding the reviewer's mention of IL/liquid crystals as reconfigurable metasurface components, as well as the previously cited comparison literature, we believe our research is unrelated because IL/liquid crystals and IL-P differ fundamentally. IL/liquid crystals are in a liquid state, whereas IL-P is a solid. This distinction fundamentally determines their physical properties and application fields, making them inherently different.

3. Beyond IL component, many active components have been investigated for tunable metasurfaces (e.g., LCs, PIN, varactors diodes, phase-change materials, mechanical deformation,). The authors should explicitly state the unique benefits of IL-based reconfigurability in comparison with these alternatives.

As the reviewer pointed out, many active components have been studied for tunable metasurfaces, and this approach can indeed be referred to as the "reconfigurable metasurfaces" mentioned by the reviewer. Coincidentally, we have already discussed the drawbacks of these schemes in the first paragraph. Our approach, however, changes the intrinsic electromagnetic properties of materials through stimulation. We emphasize once again that this represents a fundamental distinction in terms of control mechanisms and methods compared to "reconfigurable metasurfaces" (lines 57-60).

For instance, arrayed active circuit elements assembled with passive metallic structures have been investigated^{1,3-5,12}. Since external stimuli primarily affect the micron-scale circuit components within the array, the fabrication of large-area surfaces and the microwave control effects are limited.

[1] Li, F. et al. Flexible intelligent microwave metasurface with shape-guided adaptive programming. *Nat. Commun.* **16**, 3161 (2025).

[3] Qian, C. et al. Deep-learning-enabled self-adaptive microwave cloak without human intervention. *Nat. Photonics* **14**, 383-390 (2020).

[4] Lim, D. D. et al. A tunable metamaterial microwave absorber inspired by chameleon's color-changing mechanism. *Sci. Adv.* **11**, eads3499 (2025).

[5] Li, W., Xu, M., Xu, H. X., Wang, X. & Huang, W. Metamaterial Absorbers: From Tunable Surface to Structural Transformation. *Adv. Mater.* **34**, 2202509 (2022).

[12] Saifullah, Y., He, Y., Boag, A., Yang, G. M. & Xu, F. Recent Progress in Reconfigurable and Intelligent Metasurfaces: A Comprehensive Review of Tuning Mechanisms, Hardware Designs, and Applications. *Adv. Sci.* **9**, 2203747 (2022).

4. The manuscript demonstrates the proposed metasurface for stealth applications. However, in state-of-the-art stealth technology, additional factors such as broadband and continuously frequency-reconfigurable performance (not just discrete switching), operation under

oblique incidence for bi-static RADAR detection, switching speed, robustness under harsh environments (e.g., extremely low temperatures), and repeatability are all critical considerations. These aspects should be further addressed to strengthen the practical relevance of the proposed work.

Thank you for the comment. In response to the previous feedback, as well as the reviewer's repeated mention of "reconfigurable performance", we must once again emphasize that the main innovation of our paper is not based on "reconfigurable metasurfaces". In fact, we never refer to "reconfigurable metasurfaces" in our manuscript. Our work focuses on the microwave control mechanism driven by the intrinsic electromagnetic properties of materials, which are altered by external stimuli.

As illustrated in Figs. 1-3, the main focus of our research is investigating the mechanism through which the dielectric properties of ionic liquids (at the molecular level) in polymer matrices change under external (temperature) stimuli. We also explore the programmable design of dielectric properties, which depends on temperature and ionic liquid concentration, using machine learning. On this basis, we provide a case study for the programmable design of microwave absorption performance, which is one application of our dielectric control method.

In light of the main mechanisms discussed in this work, we will now address the specific questions raised by the reviewer.

- (1) Wideband and continuous frequency reconfigurable performance is indeed an important metric, but what we are discussing here is a new dynamic electromagnetic control material with a dielectric regulation mechanism, not a "reconfigurable surface." Additionally, our microwave absorption control is just an application demonstration. Based on the programmable design through machine learning, as shown in Supplementary Fig. 15, we observe that the IL-P-4 can exhibit wideband and continuous frequency reconfigurable performance in its vicinity, as shown in the following figure:

The innovation of this paper lies in the mechanistic explanation of how ionic liquids regulate dielectric properties, as well as the programmable dielectric performance and its control over microwave performance. Based on the understanding of this programmable design, the specific outcomes can include either continuous frequency-switchable performance or switchable microwave absorption performance. However, this is merely one example of the application and is not the main focus of the paper.

(2) Typically, the most direct indicator used to evaluate stealth performance is the RL (Reflection Loss) value for vertically incident electromagnetic waves, which is reflected in the main text. The operation under oblique incidence for bistatic radar detection is typically given in terms of RCS (Radar Cross Section), which is a scenario in practical applications involving the angle of incidence and the model. In Supplementary Fig. 21, we have already provided a related control conclusion with an airplane as the primary model. Relevant literature can be referenced as follows:

[1] Cheng, Z. et al. Intelligent Off/On Switchable Microwave Absorption Performance of Reduced Graphene Oxide/ VO_2 Composite Aerogel. *Adv. Funct. Mater.* 32, 2205160 (2022).

(3) The "switching speed" depends on the heating power we select, which directly affects the rate of hydrogen bond breaking and consequently controls the rate at which the electromagnetic properties of the ionic liquid polymer change. As far as we know, many temperature controllers can achieve rapid temperature control within a few seconds, such as:

[1] F. Xu. et al. Highly stretchable, fast thermal response carbon nanotube composite heater, *COMPOS PART A-APPL S.* 147 (2021) 106471.

doi.org/10.1016/j.compositesa.2021.106471.

[2] B. Zhou. et al. Ultrathin, flexible transparent Joule heater with fast response time based on single-walled carbon nanotubes/poly(vinyl alcohol) film, *Compos. Sci. Technol.* 183 (2019) 107796. doi.org/10.1016/j.compscitech.2019.107796.

(4) Regarding the "robustness and repeatability under harsh conditions (e.g., extreme low temperatures)", we have already mentioned the material preparation process. During the preparation, the material underwent 5000 high-low temperature cycles (lines 468-470).

In summary, the concerns raised by the reviewer have been addressed both in the main text and supplementary materials. We kindly ask the reviewer to verify this.

Manuscript ID: NCOMMS-25- 66869A-Z

Title: *Adaptive ionic liquid polymer microwave modulation surface with reprogrammable dielectric properties*

General Statement

We sincerely thank the editors and reviewers for their time and effort in evaluating our manuscript. We are encouraged by the reviewers' positive assessment of the substantial improvements made in the previous revision, and we greatly appreciate their constructive comments and helpful suggestions in this round, which have further guided us to refine the manuscript.

We have carefully addressed all remaining concern, with corresponding updates to the text and figures. Below, we provide a detailed, point-by-point response to each reviewer's comments. All revisions are highlighted in the revised manuscript with tracked changes.

Reviewer #1

General remark:

We thank Reviewer #1 for the positive evaluation and recommendation for acceptance.

We appreciate the reviewer's encouraging feedback and are pleased that the revisions have addressed previous concerns.

Reviewer #2

General remark:

We thank the reviewer for carefully reading our manuscript and providing valuable suggestions regarding the scope, structure, and clarify. Following these recommendations, we have substantially streamlined the excessive application-oriented discussion to emphasize the dielectric and materials-chemistry aspects of the work.

Comment 1: *The application part must be pruned since it deviates from the main aspect of the paper. This is due to the following: the authors in their reply to the authors query mention that the main focus is not microwave absorption.*

Response: We appreciate this suggestion, which has been very helpful in improving the focus and overall structure of the manuscript. The focus of this study is indeed on dielectric constants, and following the reviewer's suggestion, we have substantially shortened the application-related content in the manuscript. Currently, only the fourth subsection of the *Results* section briefly discusses the tunable microwave-absorbing surfaces enabled by dynamic dielectric control, including far-field scanning measurements and far-field RCS evaluation.

The main revisions are as follows:

- The title and content of the fifth subsection in the *Results* have been revised, and all microwave absorption-related application discussions have been moved to the Supporting Information.
- The previous analysis involving IL-P and "metasurfaces" related microwave absorption (Figures 5a-c) has been relocated to the Supporting Information, and the main-text discussion has been shortened.
- The title of the fifth subsection in the *Results* has been revised from "Metasurfaces" to "Structural Health Monitoring," which now focuses solely on the sensing performance of IL-P materials.

Deleted text (lines 410-425):

~~The metasurface structure enables absorption bandwidth control for conventional passive microwave absorbing surfaces and has achieved a breakthrough in self-sensing capability. This provides a new theoretical foundation and design approach for the development of adaptive microwave surfaces, which possess self-sensing and active stealth capabilities. We modified the tunable frequency of IL-P based active microwave absorbing surfaces by designing metasurface structures (MTS, Fig. 5a) on 3D-printed multilayer surfaces and overlaying frequency selective surfaces (FSS, Fig. 5b). As shown in Fig. 4c, the Δ EAB of IL-P-MTS and IL-P-FSS shift to lower frequencies by 2.21 GHz and 5.3 GHz, respectively, compared to IL-P-PL. This control approach extends the tunable frequency range of the IL-P based active microwave absorbing surfaces, making it advantageous for meeting the control requirements in complex electromagnetic environments. We can therefore draw a reliable conclusion that this "metasurface active surface" dual architecture offers a scalable design paradigm for next-generation intelligent camouflage skins, with frequency reconfigurability that significantly surpasses conventional control systems based on single modulation mechanisms.~~

The discussion on IL-P and MTS/FSS has been moved to the end of the third subsection 3, and the related discussion has been streamlined (lines 338-344):

The designable effective modulation bandwidth is an important requirement for reconfigurable microwave-absorbing surfaces. Firstly, impedance matching, achieved by adjusting the material thickness, can easily shift the effective modulation bandwidth (Supplementary Fig. 19). Secondly,

the metasurface structure enables the absorption bandwidth of conventional passive microwave absorbing surfaces to be modified^{1,5,12,39}. Based on this theoretical foundation and design concept, the effective modulation bandwidth of IL-P based active microwave absorbing surfaces was tailored by designing metasurface structures or overlaying frequency selective surfaces (Supplementary Fig. 20). This strategy broadens the effective modulation bandwidth, thereby improving adaptability and performance under complex electromagnetic conditions.

The original Fig. 5 has been split into Fig. 5 and Supplementary Fig. 20:

Fig. 5 | Structural Health Monitoring Enabled by IL-P Sensing Performance and CNN-Transformer Hybrid Learning Technology. **a** Sensing capability of IL-P with different IL concentrations. **b** Sensing capability of IL-P-2 with different metasurface structures. **c, d** Dynamic response curves and response/recovery times of IL-P-2-T-2. **e** Repeatability of dynamic response of IL-P-2-T-2 under 1000 Pa pressure. **f** Principle of self-sensing recognition of the IL-P-2-T-2 sensor array based on a CNN-Transformer hybrid model. **g** Confusion matrix for the classification results on the test dataset.

Supplementary Fig. 20 | Microwave Modulation Performance of IL-P and Metasurfaces. **a** Multilayer photopolymerized 3D-printed IL-P-MTS metamaterial structure. **b** IL-P-FSS structure is formed by combining a metallic disk array with IL-P. **c** RL of the metasurface IL-P in the 0.5–18 GHz.

Comment 2: *The application emphasised is the switchable microwave absorbers which is good for slow speed applications and not definitely for radar applications.*

Response: We thank the reviewer for this constructive comment and suggestion. As the reviewer mentioned, the focus of this study is on dielectric properties of the IL-P system rather than on radar absorption applications. Accordingly, we have adjusted the overall manuscript structure to reflect this focus. In the revised version, the first three figures concentrate on the dielectric properties, while only the fourth figure presents the far-field scanning and RCS of the 2mm-thick switchable microwave-absorbing surface made of IL-P. Fig. 5 now highlights the sensing performance of IL-P and its applications.

To further clarify the response behavior, we have added the temperature-time relationship of the IL-P heating process to visually evaluate the switching speed of the tunable microwave absorption. This is shown in Supplementary Fig. 22 f. The results show that a temperature change from 30°C to 130°C can be achieved within 50 s, demonstrating that the tunable microwave absorption can be effectively controlled within practical timescales.

Supplementary Fig. 22 | Pixelated microwave absorbing surface imaging. a Schematic of far-field imaging test. **b** Far-field imaging system. **c** Far-field imaging model. **d, e** Schematic of pixelated imaging control under temperature stimulation. **f** The temperature rise rate of the microwave absorbing surface with a bottom heating plate at 24V.

Comment 3: While the material does not have magnetic part, the evaluation of magnetic parameters at microwave frequencies are obtained without using the proper model meant only for dielectric materials.

Response: We thank the reviewer for raising this question. The lack of description regarding the source and application of permeability data (μ' , μ'') in the manuscript led to some confusion. In the revised manuscript, this point has been clarified.

The dielectric constants (ϵ' , ϵ'') and permeability (μ' , μ'') in the 2-18 GHz range were measured simultaneously using a vector network analyzer and coaxial measurement method. We obtained these four measured parameters— ϵ' , ϵ'' , μ' , and μ'' —across the 2-18 GHz range. From the

measurements, as expected for a non-magnetic polymer system, we observed that the dielectric constants change with temperature stimuli, while the permeability remains nearly constant.

Based on these measurements, machine learning models were employed to predict the values of these four parameters at different temperatures and ionic liquid concentrations. Using the predicted values of these four parameters, we were then able to calculate the estimated RL (reflection loss) values of a microwave-switchable surface composed of IL-P with a specific thickness (e.g., 2mm) for normal-incidence electromagnetic waves. Inverse calculations were further performed to identify the temperature and ionic liquid concentration required to achieve targeted dielectric modulation.

These clarifications have been incorporated into the revised manuscript as follows:

The lines 216-218 of the manuscript:

The permittivity (ϵ' , ϵ'') and permeability (μ' , μ'') of the material were measured at different temperatures across 2–18 GHz using a vector network analyzer (VNA) to evaluate its dielectric and magnetic responses.

The lines 304-312 of the manuscript:

Building on the aforementioned electromagnetic response mechanism, we further introduced machine learning algorithms to achieve high-precision modeling and prediction of the electromagnetic parameters of the IL-P system under different temperature conditions □ As shown in Fig. 3e and Supplementary Fig. 15. Based on the measured data (Supplementary Fig. 5, 7, 12, 13), significant differences in the variation trends of permittivity and permeability can be observed. In response to these differences, we used the EGPR model to predict ϵ' and ϵ'' , and the GBDT model to predict μ' and μ'' , thereby systematically revealing the coupled regulatory relationship between IL concentration, temperature stimulation, electromagnetic parameters, and RL.

Comment 4: *The equation for RL (S8) still has a problem since it uses $(Z_{in}-1)/(Z_{in}+1)$ while Z_{in} is not a normalized quantity.*

Response: We thank the reviewer for the careful observation. We have addressed the issues with the equations and made the necessary corrections.

$$RL(dB) = 20 \log_{10} \frac{|Z_{in}-Z_0|}{|Z_{in}+Z_0|} \quad (S8)$$

In terms of impedance matching coefficients ($Z = Z' + jZ''$, calculated by Z_{in}/Z_0 , $Z_0 = 376.73031 \Omega$), when Z is close to 1 ($Z' = 1$ and $Z'' = 0$), the EMWs can enter the absorber without reflection.

Reviewer #3

General remark:

We thank the reviewer for the careful evaluation and for the constructive comments that helped us clarify the conceptual distinction of our work and improve the presentation. The following revisions have been made accordingly.

Comment 1: *The authors argue that their work should not be categorized as a reconfigurable metasurface.*

This argument is unconvincing and inconsistent with the current understanding of reconfigurable metasurfaces.

By widely accepted definitions, a reconfigurable metasurface refers to any electromagnetic surface whose scattering or absorption properties can be dynamically altered through external stimuli, whether the underlying mechanism is:

- mechanical deformation,*
- electrical bias or carrier tuning, or*
- material-level modulation, such as temperature-driven ionic or molecular reorientation.*

In the present study, the authors clearly describe a temperature-induced modulation of dielectric properties through hydrogen-bond reconstruction and ionic motion within the IL – P matrix.

This results in measurable shifts of reflection loss and absorption bandwidth—i.e., a dynamic reconfiguration of electromagnetic response.

Such behavior squarely fits the definition of a reconfigurable metasurface, even in the absence of mechanical actuation.

Moreover, the manuscript itself repeatedly employs terminology that explicitly conveys reconfigurability—namely:

- "Adaptive ionic liquid polymer", "microwave modulation", "Reprogrammable dielectric properties"*
- "Reconstruction of hydrogen bonds", "Controllable modulation of dielectric properties at microwave frequencies", "Switchable microwave absorbing", "Tunable effective absorption"*

These expressions inherently describe stimuli-responsive electromagnetic platforms whose properties can be tuned or reprogrammed.

It is therefore inconsistent for the authors to employ this language while simultaneously claiming that their device is not a reconfigurable metasurface.

This distinction appears semantic rather than technical, and it undermines the conceptual clarity of the work.

Response: We thank the reviewer for this thoughtful comment and for outlining a broad definition of *reconfigurable metasurfaces*. We note that within the microwave and materials research communities, another widely adopted convention refers to metasurfaces as a two-dimensional periodic structure composed of subwavelength resonant unit cells, whose electromagnetic response is governed by geometry and arrangement.

Our group has previously published work following this convention, for example:

- Weihao Yang, et al. A self-biased non-reciprocal magnetic metasurface for bidirectional phase modulation. *Nature Electronics*. 6, 225 – 234 (2023)

In addition, similar definitions have been adopted in representative studies across the field:

- Alexander A. H. et al. Visible-frequency hyperbolic metasurface. *Nature*. 522, 192 – 196 (2015)
- Li, W. et al. Metamaterial Absorbers: From Tunable Surface to Structural Transformation. *Adv. Mater.* 34, 2202509 (2022).
- Lim, D. D. et al. A tunable metamaterial microwave absorber inspired by chameleon's color-changing mechanism. *Sci. Adv.* 11, eads3499 (2025).

Following this convention, our system, an intrinsically homogeneous adaptive dielectric surface whose tunability originates from molecular-level ionic motion and hydrogen-bond reconstruction, does not involve any subwavelength patterning or structural reconfiguration. To avoid potential confusion between these interpretations, we have

- Carefully revised the manuscript to consistently use the term adaptive dielectric surface in the main text.
- moved the exploratory application of IL-P integrated with metasurface concepts (previously Fig. 5) to the Supporting Information.
- Renamed the title of the fifth subsection in *results* from "*Metasurfaces*" to "*Structural Health Monitoring*".

In the revised manuscript, no in-depth metasurfaces content remains; all switchable microwave-absorbing surfaces described are homogeneous IL-P films with a thickness of 2 mm.

The deleted sections of the manuscript (lines 410-425):

~~The metasurface structure enables absorption bandwidth control for conventional passive microwave absorbing surfaces and has achieved a breakthrough in self sensing capability. This provides a new theoretical foundation and design approach for the development of adaptive microwave surfaces, which possess self sensing and active stealth capabilities. We modified the tunable frequency of IL-P based active microwave absorbing surfaces by designing metasurface structures (MTS, Fig. 5a) on 3D printed multilayer surfaces and overlaying frequency selective surfaces (FSS, Fig. 5b). As shown in Fig. 4c, the Δ EAB of IL-P-MTS and IL-P-FSS shift to lower frequencies by 2.21 GHz and 5.3 GHz, respectively, compared to IL-P-PL. This control approach extends the tunable frequency range of the IL-P based active microwave absorbing surfaces, making it advantageous for meeting the control requirements in complex electromagnetic environments. We can therefore draw a reliable conclusion that this "metasurface active surface" dual architecture offers a scalable design paradigm for next generation intelligent camouflage skins, with frequency reconfigurability that significantly surpasses conventional control systems based on single modulation mechanisms.~~

In the revised manuscript, the discussion of IL-P in conjunction with "metasurfaces" has been moved to the end of the third subsection of the Results section, and the "metasurface" related discussion has been streamlined (lines 338-344):

The designable effective modulation bandwidth is an important requirement for reconfigurable microwave-absorbing surfaces. Firstly, impedance matching, achieved by adjusting the material thickness, can easily shift the effective modulation bandwidth (Supplementary Fig. 19). Secondly, the metasurface structure enables the absorption bandwidth of conventional passive microwave absorbing surfaces to be modified^{1,5,12,39}. Based on this theoretical foundation and design concept, the effective modulation bandwidth of IL-P based active microwave absorbing surfaces was tailored by designing metasurface structures or overlaying frequency selective surfaces (Supplementary Fig. 20). This strategy broadens the effective modulation bandwidth, thereby improving adaptability and performance under complex electromagnetic conditions.

The previous version of Fig. 5 has been split into Fig. 5 and Supplementary Fig. 20:

Fig. 5 | Structural Health Monitoring Enabled by IL-P Sensing Performance and CNN-Transformer Hybrid Learning Technology. **a** Sensing capability of IL-P with different IL concentrations. **b** Sensing capability of IL-P-2 with different metasurface structures. **c, d** Dynamic response curves and response/recovery times of IL-P-2-T-2. **e** Repeatability of dynamic response of IL-P-2-T-2 under 1000 Pa pressure. **f** Principle of self-sensing recognition of the IL-P-2-T-2 sensor array based on a CNN-Transformer hybrid model. **g** Confusion matrix for the classification results on the test dataset.

Supplementary Fig. 20 | Microwave Modulation Performance of IL-P and Metasurfaces **a** Multilayer photopolymerized 3D-printed IL-P-MTS metamaterial structure. **b** IL-P-FSS structure is formed by combining a metallic disk array with IL-P. **c** RL of the metasurface IL-P in the 0.5–18 GHz.

Comment 2: *The authors' response regarding liquid crystal (LC)-based tunable metasurfaces does not sufficiently address the reviewer's concern.*

The question was not about the chemical phase (liquid vs. solid), but about the functional analogy between IL - P and LC systems.

Both rely on orientation-dependent polarization and external-stimulus-driven molecular or ionic reordering to achieve dielectric and electromagnetic tunability.

Thus, their operating principles are physically comparable.

To substantiate the claimed novelty of the IL - P approach, the authors should provide a quantitative or conceptual comparison with LC-based metasurfaces, such as:

- *the range and reversibility of dielectric tunability ($\Delta \epsilon'$, $\Delta \epsilon''$),*
- *response time and operational stability,*
- *frequency range (microwave vs. optical), and*
- *fabrication scalability or robustness.*

Without such analysis, the distinction between IL - P and LC systems remains superficial, and the claimed advantage of the IL - P-based approach is not convincingly demonstrated.

Response: We thank the reviewer for highlighting this point. We have included a comparison of relevant literature in the supporting materials, including typical cases of liquid crystals (LCs) and other related studies. The comparison summarizes differences in frequency range, tunability

mechanism, and operational properties between LC-based metasurfaces and the IL-P system.

In particular:

- LC-based metasurfaces achieve tunability primarily in optical frequencies (hundreds of THz) via field-induced molecular reorientation, whereas IL-P materials operate in microwave frequencies (GHz) through hydrogen-bond-driven ionic mobility.
- IL-P systems exhibit broader dielectric-modulation amplitude ($\Delta\epsilon' \approx 2-5$ at 10 GHz), high reversibility, and solid-state stability suitable for large-area fabrication.

These distinctions demonstrate that the IL-P mechanism is physically and functionally different from LC-based control. And we have added a corresponding discussion in the main text.

Supplementary Tab. 1 Comparison of IL-P with other tunable dielectric surfaces.

Type	Material	State	Frequency	ϵ' (10GHz)	ϵ'' (10GHz)	Method	Mechanism	Ref.
Liquid Crystal Metasurfaces	E7	Liquid	Light 500-900 nm	–	–	Temperature	Molecular arrangement	10
Liquid Crystal Metasurfaces	E7	Liquid	Light 580-700 nm	–	–	Voltage	Molecular arrangement	11
Liquid Crystal Metasurfaces	E7	Liquid	Light 1500-1750 nm	–	–	Voltage	Molecular arrangement	12
Active metallic metasurface	Varicap diode	Solid	Microwave 2.6-4.0 GHz	–	–	Voltage	Reconfigurable metasurfaces	13
Active metallic metasurface	Varicap diode	Solid	Microwave 8.4 GHz	–	–	Voltage	Reconfigurable metasurfaces	14
Active metallic metasurface	Varicap diode	Solid	Microwave 6-22 GHz	–	–	Voltage	Reconfigurable metasurfaces	15
Dynamic metasurface	Dynamic array	Solid	Microwave 4-12 GHz	–	–	Deformation	Reconfigurable metasurfaces	16
Metal-oxide composit foam	RGO/VO ₂ Aerogel	Solid	Microwave 1-18 GHz	10	3 to 5	Temperature	Conductivity	17
Ionic liquid polymer	IL-P-2 IL-P-6	Solid	Microwave 2-18 GHz	5.1 to 7.3 8.1 to 12.4	1.1 to 3.2 3.6 to 8.6	Temperature	Primary: Ionic conductivity Secondary: Molecular orientation movement	This work

- 11 Zou, C. et al. Electrically Tunable Transparent Displays for Visible Light Based on Dielectric Metasurfaces. *ACS Photonics* **6**, 1533-1540 (2019).
- 12 Komar, A. et al. Electrically tunable all-dielectric optical metasurfaces based on liquid crystals. *Appl. Phys. Lett.* **110**, 071109 (2017).
- 13 Li, F. et al. Flexible intelligent microwave metasurface with shape-guided adaptive programming. *Nat. Commun.* **16**, 3161 (2025).
- 14 Qian, C. et al. Deep-learning-enabled self-adaptive microwave cloak without human intervention. *Nat. Photonics* **14**, 383-390 (2020).
- 15 Li, W., Xu, M., Xu, H. X., Wang, X. & Huang, W. Metamaterial Absorbers: From Tunable Surface to Structural Transformation. *Adv. Mater.* **34**, 2202509 (2022).
- 16 Lim, D. D. et al. A tunable metamaterial microwave absorber inspired by chameleon's color-changing mechanism. *Sci. Adv.* **11**, 3499 (2025).
- 17 Cheng, Z. et al. Intelligent Off/On Switchable Microwave Absorption Performance of Reduced Graphene Oxide/VO₂ Composite Aerogel. *Adv. Funct. Mater.* **32**, 2205160 (2022).

The manuscript has been revised as follows (lines 347-350):

By controlling the IL in IL-P at the molecular level, dynamic dielectric properties in the microwave frequency range are achieved, showing superior modulation capabilities compared to existing methods, as shown in Supplementary Table 1. Building on this, the active microwave-absorbing surfaces composed of IL-P surpass most of the current reports^{2,4,6,19-21}.

Closing Note

We once again thank the reviewers and the editorial team for their valuable time and constructive feedback. We believe that the revisions and clarifications provided have fully resolved the remaining issues and significantly strengthened the manuscript. We respectfully request that the revised version be reconsidered for publication in *Nature Communications*.